# Stem cells and fluid flow drive cyst formation in an invertebrate excretory organ

Hanh Thi-Kim Vu[1,2], Jochen C Rink[3], Sean A McKinney[1], Melainia McClain[1], Naharajan Lakshmanaperumal[3], Richard Alexander[1], Alejandro Sánchez Alvarado[1,2,4]*

[1]Stowers Institute for Medical Research, Kansas City, United States; [2]Department of Neurobiology and Anatomy, University of Utah, Salt Lake City, United States; [3]Max Planck Institute of Molecular Cell Biology and Genetics, Dresden, Germany; [4]Howard Hughes Medical Institute, Stowers Institute for Medical Research, Kansas City, United States

**Abstract** Cystic kidney diseases (CKDs) affect millions of people worldwide. The defining pathological features are fluid-filled cysts developing from nephric tubules due to defective flow sensing, cell proliferation and differentiation. The underlying molecular mechanisms, however, remain poorly understood, and the derived excretory systems of established invertebrate models (*Caenorhabditis elegans* and *Drosophila melanogaster*) are unsuitable to model CKDs. Systematic structure/function comparisons revealed that the combination of ultrafiltration and flow-associated filtrate modification that is central to CKD etiology is remarkably conserved between the planarian excretory system and the vertebrate nephron. Consistently, both RNA-mediated genetic interference (RNAi) of planarian orthologues of human CKD genes and inhibition of tubule flow led to tubular cystogenesis that share many features with vertebrate CKDs, suggesting deep mechanistic conservation. Our results demonstrate a common evolutionary origin of animal excretory systems and establish planarians as a novel and experimentally accessible invertebrate model for the study of human kidney pathologies.

*For correspondence: asa@stowers.org

**Reviewing editor**: Yukiko M Yamashita, University of Michigan, United States

## Introduction

The vertebrate kidney plays a pivotal role in the maintenance of organismal homeostasis in the face of changing external and internal conditions. Its myriad individual functions, including the removal of metabolic waste products, regulation of ion concentrations and acid/base balance, are all tied to two basic physiological processes: (1) the pressure-driven ultra-filtration of blood plasma across the glomerulus, whereby molecular sieves prevent the passage of large macromolecules (e.g., plasma proteins); and (2) the subsequent modification of the resulting filtrate during its passage through the epithelial nephron tube (*Ruppert and Smith, 1988*; *Ruppert, 1994*). The parallel operation of many millions of glomerulus/nephron units allows formidable filtration rates, amounting to 170 liters of primary filtrate/day in a healthy human adult. In line with the pivotal homeostatic roles of the kidney, kidney diseases pose a serious health problem. The most common human kidney disorders are cystic kidney diseases (CKDs), affecting nearly 12 million people worldwide (*Priolo and Henske, 2013*). CKDs encompass a wide range of hereditary, developmental, and acquired conditions (*Bisceglia et al., 2006*), all of which share the pathological hallmark of fluid-filled cysts developing in the kidney. This has led to the suggestion that the molecular mechanisms causing cyst formation are similar, or at least, share a common pathway (*Watnick and Germino, 2003*). The molecular cloning of multiple

**eLife digest** Millions of people around the world are affected by cystic kidney diseases, which are amongst the most common inherited genetic disorders. Throughout their life, people with these diseases develop fluid-filled cysts in their kidneys, which stop these organs from working properly and can eventually lead to organ failure.

Healthy kidneys perform many vital roles in the body, including removing waste products and keeping the concentration of salts in the blood in balance. These activities depend on the kidneys filtering the blood, and then reabsorbing useful chemicals from the filtered fluid as it passes through small tubes called tubules. Cysts disrupt both of these processes.

Mutations in many different genes can cause cystic kidney diseases. Many of these genes encode proteins that are involved in the formation of cilia: hair-like structures that project from some cell membranes. Cilia on the cells that line tubules are thought to bend in response to the flow of fluid and then generate signals that dampen cell proliferation. This would explain how the loss of cilia could cause too many cells to develop, which would lead to the formation of cysts. But many of the molecular details are missing from this explanation. Previous studies in mammals and simple model organisms (such as fruit flies and roundworms) have sought to fill in the gaps, but each model has its own limitations.

Now, Thi-Kim Vu et al. propose that a flat worm called a planarian could provide a new and experimentally accessible animal model to study cystic kidney diseases. These flat worms get rid of their waste products via an excretory system that consists of branched tubules that spread throughout the body. Thi-Kim Vu et al. found that, like the tubules in the kidney, these tubules filter and then reabsorb chemicals from body fluids. Moreover, these processes are performed in different parts of the tubules, exactly as they are in the tubules in kidneys.

The genome of a flat worm called *Schmidtea mediterranea* contains many genes that cause cysts to form when their equivalents are mutated in humans. Reducing the expression of these genes (and others that are involved in cilia formation) also caused cysts to form in the flat worms.

These findings indicate that it is likely that the excretory systems of different animals have a shared evolutionary history. If so, the findings support the idea that cilia in kidney tubules send signals in response to fluid flow that affect kidney-specific stem cells. They also suggest that problems with these signals could be at the core of some human cystic kidney diseases. One of the next challenges will be to identify these cilia-associated signals. Finally, given that studies involving thousands of flat worms can be carried out with minimal cost, the ultimate goal is to develop flat worms into a new model to discover and investigate genes linked to human kidney diseases.

CKD mutations and the realization that the affected genes all function at the primary cilia, basal bodies or centrosomes, has given rise to the ciliary hypothesis as a unifying disease mechanism of CKDs (*Yoder et al., 2002*; *Mollet et al., 2005*; *Fliegauf et al., 2006*). Accordingly, the primary cilia of tubule cells are thought to act as flow sensors, eliciting intracellular calcium fluxes through stretch sensitive polycystin channels in response to flow-driven bending (*Praetorius and Spring, 2001, 2003*; *Nauli et al., 2003*; *Praetorius et al., 2004*). These signals are thought to constitutively dampen cell proliferation, such that loss of filtrate flow or interruptions in the signal transduction process precipitate chronic overproliferation and consequently cyst formation (*Deane and Ricardo, 2012*). However, major mechanistic aspects of the ciliary hypothesis remain poorly understood, including the integration of the calcium signal with downstream transcriptional regulation of cell behavior (*Wilson and Goilav, 2007*; *Uhlenhaut and Treier, 2008*; *Deane and Ricardo, 2012*; *Kotsis et al., 2013*), the extent by which cyst development can be understood as chronic activity of endogenous repair mechanisms (*Deane and Ricardo, 2012*), and the identity and origins of the ectopically over-proliferating cells (*Murer et al., 2002*; *Weimbs, 2007*; *Lodi et al., 2012*). Further, these questions present an investigative challenge, given the poor experimental accessibility of the mammalian kidney as an internal and essential organ. The *Xenopus* pronephros and zebrafish pro- and mesonephric kidneys, therefore, are increasingly being explored as model systems for human kidney disease (*Drummond, 2005*; *Ebarasi et al., 2011*). Compounding this problem is the fact that it has not been possible to bring the full power of invertebrate models in solving fundamental cell biological

processes to the analysis of human kidney disease (*Igarashi, 2005*; *Dow and Romero, 2010*). Both *Caenorhabditis elegans* and *Drosophila melanogaster* have highly derived excretory organs in which ultrafiltration is either entirely lacking (*C. elegans*; [*Buechner, 2002*]) or uncoupled from reabsorption/ secretion (*D. melanogaster*; [*Dow and Romero, 2010*]). Furthermore, the excretory cells of both organisms are lacking cilia as a further requirement for modeling CKDs. However, *C. elegans* or *Drosophila* are but two of the myriad invertebrate species and multiple studies have documented the existence of more complex excretory systems outside the *Ecdysozoa* (*Ruppert and Smith, 1988*). One such example is the excretory system of planarian flatworms. We and others have previously reported on intriguing similarities between planarian protonephridia and the vertebrate nephron (*Rink et al., 2011*; *Scimone et al., 2011*). Here, we carried out a systematic structure function comparison to systematically gauge the potential of planarian protonephridia as a model system for human kidney diseases. Our results demonstrate the structural coupling of cilia-driven ultrafiltration and filtrate modification in planarian protonephridia, as well as extensive topological homology of solute carrier expression domains with the vertebrate nephron. These structure/function homologies extend to common pathologies, including shared requirements of nephrin in the maintenance of the ultrafiltration barrier, and of nephrocystins in preventing the development of tubular cysts. Our results therefore establish planarian protonephridia as a novel and viable invertebrate model for studying human kidney development and diseases.

## Results

### Protonephridia are ultrafiltration devices in planarian

The planarian excretory system consists of branched epithelial tubules (protonephridia) distributed throughout the entire body plan (*Figure 1A*) (*Rink et al., 2011*). The barrel-shaped flame cells capping the proximal tubule (PT) ends have been proposed to act as unicellular ultrafiltration devices solely on the basis of morphological evidence (*Figure 1B*) (*Wilhelmi, 1906*; *Wilson and Webster, 1974*). To functionally test this premise, we adapted an assay previously used to demonstrate the ultrafiltration capacity of *Drosophila* nephrocytes (*Weavers et al., 2009*; *Zhuang et al., 2009*). We co-injected two inert and differentially labeled tracer molecules of different sizes into the anterior planarian mesenchyme (10 kDa and 500 kDa molecular weight dextrans). Already at 2-hr post injection, we found robust tracer accumulation in protonephridia throughout the body, confirming their active role in extracellular fluid processing. Interestingly, only the small molecular weight tracer produced intense and continuous protonephridial labeling, whereas the large dextran displayed weak and patchy labeling (*Figure 1C,D*). Because the two tracer molecules in the injection mix carried equal numbers of fluorophores, the preferential accumulation of the small over the large dextran particles indicates molecular size filtration upon entry into the protonephridial system. We conclude from these experiments that planarian protonephridia, like vertebrate nephrons, combine ultrafiltration with filtrate modification in the same structure.

### Unexpected complexity of protonephridial tubules

We next sought to investigate the filtrate modification capacities of the planarian protonephridial system. In the vertebrate nephron, the expression of a large number and diverse types of solute carrier (slc) transporters recovers essential molecules from the primary filtrate or secretes waste products into the tubule lumen (*Landowski, 2008*; *Raciti et al., 2008*). The known substrate specificity of slc families together with their restricted expression in specific nephron segments establishes a structure/function topology of filtrate modification processes along the nephron. Towards the dual goal of identifying and mapping solute modification processes in planarian protonephridia, we set out to identify, clone and expression-map all solute carriers in the planarian genome. A systematic sequence homology search of the planarian *Schmidtea mediterranea* genome identified 318 *slc* genes. Reciprocal BLAST analysis and sequence alignments revealed that *S. mediterranea* slcs represent 43 slc families (*Figure 2—figure supplement 1*, *Figure 2—figure supplement 2*, *Figure 2—figure supplement 3*, *Figure 2—figure supplement 4*, *Figure 2—figure supplement 5*, *Figure 2—figure supplement 6*, *Figure 2—figure supplement 7*, *Supplementary file 1*). Expression patterns of all *slc* genes were analyzed by in situ hybridization in intact asexual planarians. We obtained expression patterns for 287 genes in various tissues (*Figure 2—figure supplement 8*, *Figure 2—figure supplement 9*, *Figure 2—figure supplement 10*, *Figure 2—figure supplement 11*, *Figure 2—figure supplement 12*,

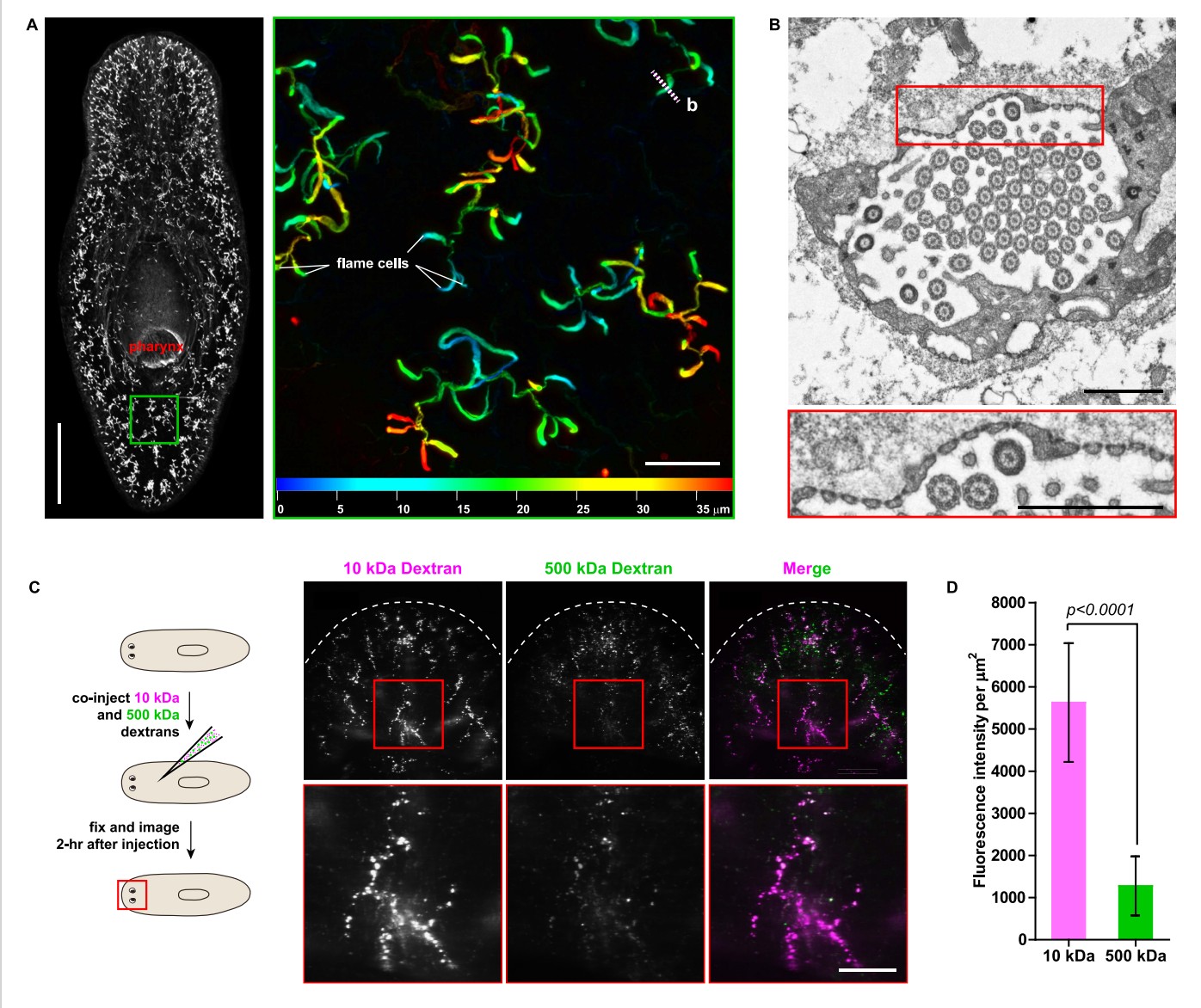

**Figure 1**. Protonephridia are ultrafiltration devices in planarians. (**A**) Whole-mount acetylated tubulin (AcTub) staining. Scale bars: 500 µm. Inset shows depth-coded projection of AcTub staining. Superficial structures are in blue and deeper structures are in red. Scale bars: 50 µm. (**B**) Cross-section through a flame cell. Inset shows a high magnification of filtration diaphragm. Scale bar: 1 µm. (**C**, **D**) Ultrafiltration assay assessing ultrafiltration capacity in the planarian protonephridia. (**C**) Fluorescent overlay showing dextran uptake in the animals that co-injected with 10 kDa and 500 kDa fluorescently labeled dextran. Inset showing a high magnification of tubule structure labeled by dextran. Scale bar: 100 µm. (**D**) Quantification of small and large dextran uptake.

*Figure 2—figure supplement 13*), with 49 of these displaying putative protonephridial expressions. The expression of such a large fraction of *slc* genes in protonephridial tubules already indicated a rich potential for solute modifications.

In order to establish a comprehensive structure-function map of protonephridia, we next mapped the expression domain of each protonephridial *slc* relative to two previously characterized markers (*Figure 2A*, top; *Supplementary file 2*): (1) acetylated tubulin (AcTub) antibody staining, which marks flame cells and the adjoining PT segment; and (2) *Smed-CAVII-1*, which is expressed in the adjacent distal tubule (DT) segment (*Rink et al., 2011*). Markers for the domain distal to *CAVII-1* expression were not available at the beginning of this study. Fluorescent in situ hybridization (FISH) mapping of putative protonephridial *slc* genes against the two markers and general tubule anatomy

(e.g., branched vs coiled PT segments) revealed a significantly greater complexity of protonephridial cell types than previously appreciated (*Figure 2A*; *Figure 2—figure supplement 14*, *Figure 2—figure supplement 15*, *Figure 2—figure supplement 16*, *Figure 2—figure supplement 17*, *Figure 2—figure supplement 18*, *Figure 2—figure supplement 19*, *Figure 2—figure supplement 20*). *slc* expression domains define at least three sub-domains within the PT (PT1, PT2, and PT3; *Figure 2A–D*) and the non-overlapping expression of representative *slc* genes in 3-color FISH experiments demonstrates the significance of the inferred PT subdivisions (*Figure 2—figure supplement 14*, *Figure 2—figure supplement 15*, *Figure 2—figure supplement 16*, *Figure 2—figure supplement 17*). Similarly, we found that *slc* expression domains divide the DT into 2 sub-domains (DT1 and DT2; *Figure 2A,D–F*; *Figure 2—figure supplement 18F*). Interestingly, the *slc12a-4* expression domain extended beyond *CAVII-1* expression, where it was co-expressed with a further cohort of 14 *slc* genes, including *Smed-slc24a-3* (*Figure 2A,G*, *Figure 2—figure supplement 19*, [*Scimone et al., 2011*]). Together, these 14 *slc* genes therefore define the so far unknown continuation of protonephridia beyond the *CAVII-1* expression domain, which for reasons detailed below we refer to as the 'Collecting Duct' (CD). Interestingly, CD marker expressing segments were exclusively located close to the dorsal body surface, supporting early reports suggesting that the protonephridial terminus was located in the dorsal epithelium (*Wilhelmi, 1906*). Consistently, sagittal sections revealed occasional CD segments crossing the basement membrane and appearing to terminate in the single-layered outer epithelium (e.g., *Smed-slc12a-1*, *Figure 2H*). To confirm this finding, we performed electron microscopy (EM) on serial thin sections and succeeded in visualizing multiple examples of ducts connecting into the dorsal epithelium and opening directly to the exterior (*Figure 2I*, *Video 1*). The presence of mitochondria and numerous small vesicles is ultrastructural characteristics of this region, similar to that of type B intercalated cells in the vertebrate CD.

For the first time, our results trace the complete course of protonephridial tubules from the ultrafiltrating flame cells as proximal entry point to their terminus in the dorsal epithelium. Further, our systematic mapping of expression domains of *slc* genes defined 6 molecularly distinct segments along the proximal-distal axis of protonephridia.

## Extensive functional homology between planarian protonephridia and vertebrate nephrons

We next took advantage of our expression data and the known transport activities of slc families in vertebrates to infer possible functional specializations of these 6 protonephridial segments. Clustering a subset of *slc* genes with known substrate specificity by substrate class and site of expression revealed a striking segregation of similar transport activities into similar regions of the protonephridial tubule, indicating the functional specialization of different segments (*Figure 3A*, top). Because this subset of *slc* genes was intentionally chosen due to its known representation for transport activities of specific segments of the nephron (*Raciti et al., 2008*), this map afforded a basis for direct structure/ function comparisons with the vertebrate nephron. Constructing a similar map of *slc* expression in the rodent metanephros based on published data (*Figure 3A*, bottom; *Supplementary file 3*) revealed striking parallel: Not only is the sequence of slc family expression very similar along the filtrate flow axis, but almost all nephron segments have clearly identifiable homologous segments in protonephridia. In vertebrates, the PT is responsible for the reabsorption of more than 70% of filtered solutes from the primary urine, including inorganic/organic ions and vital nutrients (glucose, amino acids, and vitamins). The homologous *slc* expression of planarian PT1-3 and the preferential labeling of PT1-2 by injected dextran (*Figure 3B*) provide strong evidence that the proximal protonephridial segments are likewise primarily responsible for the recovery of filtered substances. The DT plays an important role in acid-base homeostasis by reabsorbing bicarbonates and secreting protons into the urine (*Carraro-Lacroix and Malnic, 2010*). The corresponding expression of bicarbonate (e.g., *Smed-slc4a-6*, *Figure 2—figure supplement 18A*) or proton transporters (e.g., $Na^+/H^+$ exchanger *Smed-slc9a-3*, *Figure 2E*) in DT1 and DT2 suggests a similar function of these protonephridial segments. Consistently, the RNAi knockdown of *slc4a-6* caused a measurable acidification of the intercellular milieu (*Figure 3C*), supporting a functionally conserved role of DT1-2 in planarian pH homeostasis. Moreover, the vertebrate CD comprises distinct cortical and medullary segments and mediates the bulk of water recovery/urine concentration (*Nielsen et al., 2002*). The shared expression of the bicarbonate transporter *Smed-slc4a-7* and the ammonia transporter

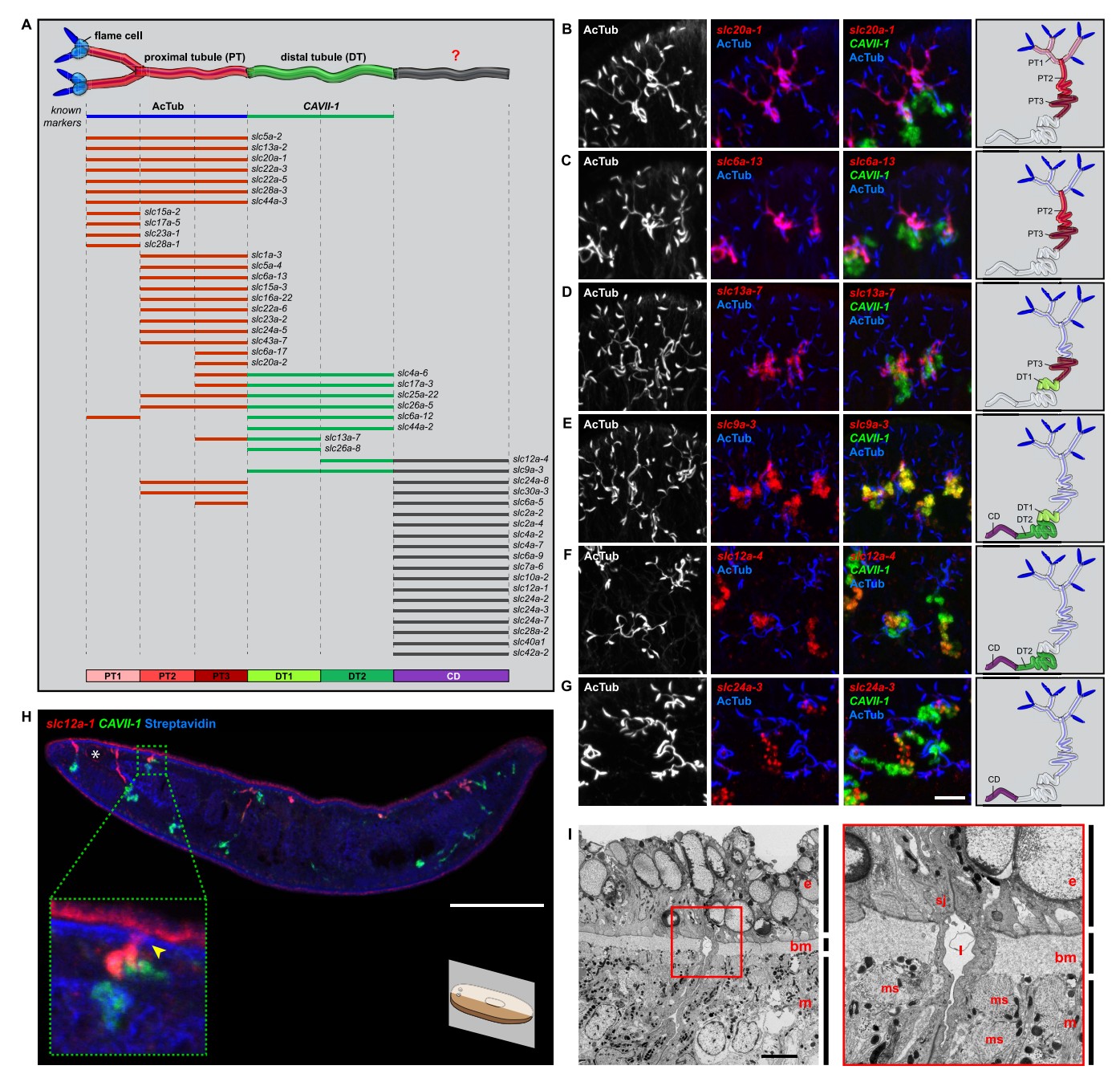

**Figure 2**. Unexpected complexity of protonephridial tubules is revealed by systematic gene expression mapping of slc genes along the protonephridial tubules. (**A**) Cartoon shows previous segmentation model of the protonephridial tubule and expression map of *slc* genes along the protonephridial tubule. (**B–G**) Representative images show expression domains of selected *slc* genes in (**B**) PT1, PT2 and PT3, (**C**) PT2 and PT3, (**D**) PT3, (**E**) DT1, DT2 and CD, (**F**) DT2 and CD and (**G**) CD. Fluorescent overlay of the indicated gene (red) with PT marker (AcTub) and distal tubule (DT) marker (*CAVII-1*). A color-coded scheme of the protonephridial tubule at the end of each panel represents the expression domain of the indicated gene. Images are maximum projections of confocal Z-sections. Scale bars: 50 µm. (**H**) Longitudinal-section through a worm shows a dorsal-bias expression of *slc12a-1*. Fluorescent overlay of *slc12a-1* with DT marker (*CAVII-1*) and streptavidin (which labels the basement membrane of several planarian epithelial structures, including the outer epithelium). Inset shows a magnification of CD, visualized by *slc12a-1*, crossing the basement membrane of the dorsal epithelia. Yellow arrowhead, exterior opening of the CD. Scale bars: 200 µm. (**I**) TEM image shows CD connected to the dorsal epithelia. Inset shows a magnification of CD connected to the dorsal epithelia. e, epithelia; bm, basement membrane; m, mesenchyme; sj, septate junction; l, lumen; ms, muscle. Scale bars: 5 µm.

The following figure supplements are available for figure 2:

*Figure 2. continued on next page*

*Figure 2. Continued*

**Figure supplement 1**. Solute carrier gene families in the planarian *Schmidtea mediterranea*.

**Figure supplement 2**. Schematic representation of phylogenetic clusters of γ- (**A**), δ- (**B**) groups of *slcs* and the Tim barrel- (**C**), IT- (**D**), Drug/Metabolite (**E**) transporter clans of *slcs*.

**Figure supplement 3**. Schematic representation of phylogenetic clusters of α-groups of *slcs*.

**Figure supplement 4**. Schematic representation of phylogenetic clusters of β-groups of *slcs*.

**Figure supplement 5**. Schematic representation of phylogenetic clusters of Smed-slc1a (**A**), Smed-slc5a (**B**), Smed-slc22a (**C**), Smed-slc6a (**D**).

**Figure supplement 6**. Schematic representation of phylogenetic clusters of Smed-slc4a (**A**), Smed-slc7a (**B**), Smed-slc12 (**C**), Smed-slc15 (**D**), Smed-slc20 (**E**), Smed-slc23 (**F**), Smed-slc26 (**G**).

**Figure supplement 7**. Schematic representation of phylogenetic clusters of Smed-slc28 (**A**), Smed-slc30 (**B**), and Smed-slc42 (**C**).

**Figure supplement 8**. Expression patterns of *slc* genes that belong to solute carrier families 1–6 in an asexual strain of the planarian *S. mediterranea*.

**Figure supplement 9**. Expression patterns of *slc* genes that belong to solute carrier families 7–15 in an asexual strain of the planarian *S. mediterranea*.

**Figure supplement 10**. Expression patterns of *slc* genes that belong to solute carrier families 16–23 in an asexual strain of the planarian *S. mediterranea*.

**Figure supplement 11**. Expression patterns of *slc* genes that belong to solute carrier families 24–29 in an asexual strain of the planarian *S. mediterranea*.

**Figure supplement 12**. Expression patterns of *slc* genes that belong to solute carrier families 30–38 in an asexual strain of the planarian *S. mediterranea*.

**Figure supplement 13**. Expression patterns of *slc* genes that belong to solute carrier families 40–47 in an asexual strain of the planarian *S. mediterranea*.

**Figure supplement 14**. Expression of *slc* genes in the PT.

**Figure supplement 15**. Expression of *slc* genes in the PT1 segment of the PT.

**Figure supplement 16**. Expression of *slc* genes in the PT2 and PT3 segments of the PT.

**Figure supplement 17**. Expression of *slc* genes in PT3 segment of the PT.

**Figure supplement 18**. Expression of *slc* genes in the DT.

**Figure supplement 19**. Expression of *slc* genes in the collecting duct.

**Figure supplement 20**. Expression of *slc* genes weakly expressed in both proximal and DTs.

*Smed-slc42a-2* in the terminal segment (*Figure 3A*) supports a basal homology between the CD and the corresponding protonephridial segment, which is why we have chosen to adopt the vertebrate nomenclature. However, the large number of additional *slc* genes expressed in the protonephridial CD (*Figure 2A*) and the lack of aquaporin expression (not shown) suggest divergent functions.

The only nephron segment for which our analysis did not identify a protonephridial homologue was the intermediate tubule (IT). In terrestrial vertebrates, IT and CD have tightly linked functions in water conservation, whereby urea secretion by the IT establishes high extracellular solute concentrations that aid in water reabsorption from the CD (*Pannabecker, 2012*). As freshwater animals, planarian protonephridia have to clear, rather than conserve water, providing a compelling rationale for why specifically IT and CD are divergent. Such functional diversity of IT/CD segments is also observed in the pronephric kidneys of freshwater vertebrates, such as zebrafish (*Wingert and Davidson, 2008*).

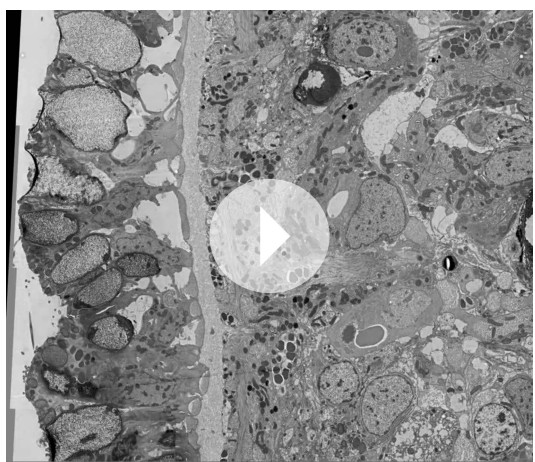

**Video 1.** Protonephridial collecting duct opens to the dorsal epithelia. Serial TEM images showing the protonephridial collecting duct connected to the dorsal epithelia.

Altogether, our analyses reveal a striking structural and functional homology between the vertebrate nephron and the planarian protonephridium.

## Recapitulation of podocyte slitdiaphragm pathologies in flame cells

We next asked whether the homologies between nephrons and protonephridia extend to common pathologies. The striking structural similarities between the ultrafiltration sites in the two systems, podocyte foot processes (*Pavenstadt et al., 2003*) and flame cell filtration barriers (*Figure 1B*) could reflect a requirement for common components. In humans, mutations in the large IgG-repeat transmembrane proteins *NPHS1* and *NEPH1* cause slit diaphragm loss by fusion of neighboring foot processes into a continuous cytoplasmic sheet (foot process effacement), resulting in proteinuria and edema (*Kestila et al., 1998*; *Donoviel et al., 2001*). Systematic sequence homology searches of the *S. mediterranea* genome identified 7 *NPHS1* homologs and 3 *NEPH* homologs (*Figure 4—figure supplement 1*). Interestingly, *Smed-NPHS1-6* and *Smed-NEPH-3* were expressed in flame cells (*Figure 4A,B*) and RNAi of both genes produced strong bloating and partial clearing of body pigmentation (*Figure 3C*). Both phenotypes have previously been identified as characteristic hallmarks of tissue edema (*Rink et al., 2011*), thus providing a strong indication that the genes are required for the function of the planarian excretory system.

Because flame cell numbers appeared normal in both intact and regenerating animals (*Figure 4—figure supplement 2*), we examined the ultrastructure of the filtration diaphragm in *NPHS1-6* and *NEPH-3(RNAi)* planarians. Wild type flame cells display slit-shaped 35–40 nm wide fenestrae that form between 90–150 nm wide foot processes (*Figures 1B*, *4D*). Under RNAi knockdown of either *NPHS1-6* or *NEPH-3*, the filtration diaphragm was completely absent and the foot processes underwent apparent effacement in both intact (*Figure 4D*, *Videos 2*, *3*) and regenerating animals (*Figure 4—figure supplement 3*). The dextran injection assay confirmed the loss of ultrafiltration capability in *NPHS1-6(RNAi)* planarians, which displayed equal uptake of both small and large molecular tracer in the PT (*Figure 4E,F*). Why would the fusion of foot processes into a continuous sheet result in loss of filtration size selectively, rather than a general block of filtration? In human nephrotic syndrome patients, the loss of ultrafiltration capability in thought to occur as a consequence of podocyte detachment or apoptosis and subsequent filtrate leakage (*Tojo and Kinugasa, 2012*). However, many nephrotic syndrome patients present ultrafiltration deficiencies without podocyte detachment or loss (*Furness et al., 1999*; *Lahdenkari et al., 2004*). Consistently, in *NPHS1-6-* and *NEPH-3(RNAi)* planarians, we could not observe ultrastructural evidence of flame cell detachment at the time that loss of ultrafiltration was observed (not shown). Evidence from a rat model of nephrotic syndrome indicates that the upregulation of transcytotic transport processes across the effaced podocyte envelope could maintain a basal level of non-size-selective fluid flow (*Tojo et al., 2008*; *Kinugasa et al., 2011*; *Tojo and Kinugasa, 2012*), yet the exact mechanisms remain to be determined in both humans and planarians. Regardless, the striking parallels between the *NPHS1-6-* and *NEPH-3(RNAi)* phenotypes in planarians and human nephrotic syndrome demonstrate that the functional homology between planarian flame cells and vertebrate podocytes extends to common pathologies.

## Cyst formation in planarian PTs

Encouraged by these results, we expanded our search for conserved pathologies to the protonephridial tubules. The most common class of human inherited disorders affecting the nephron

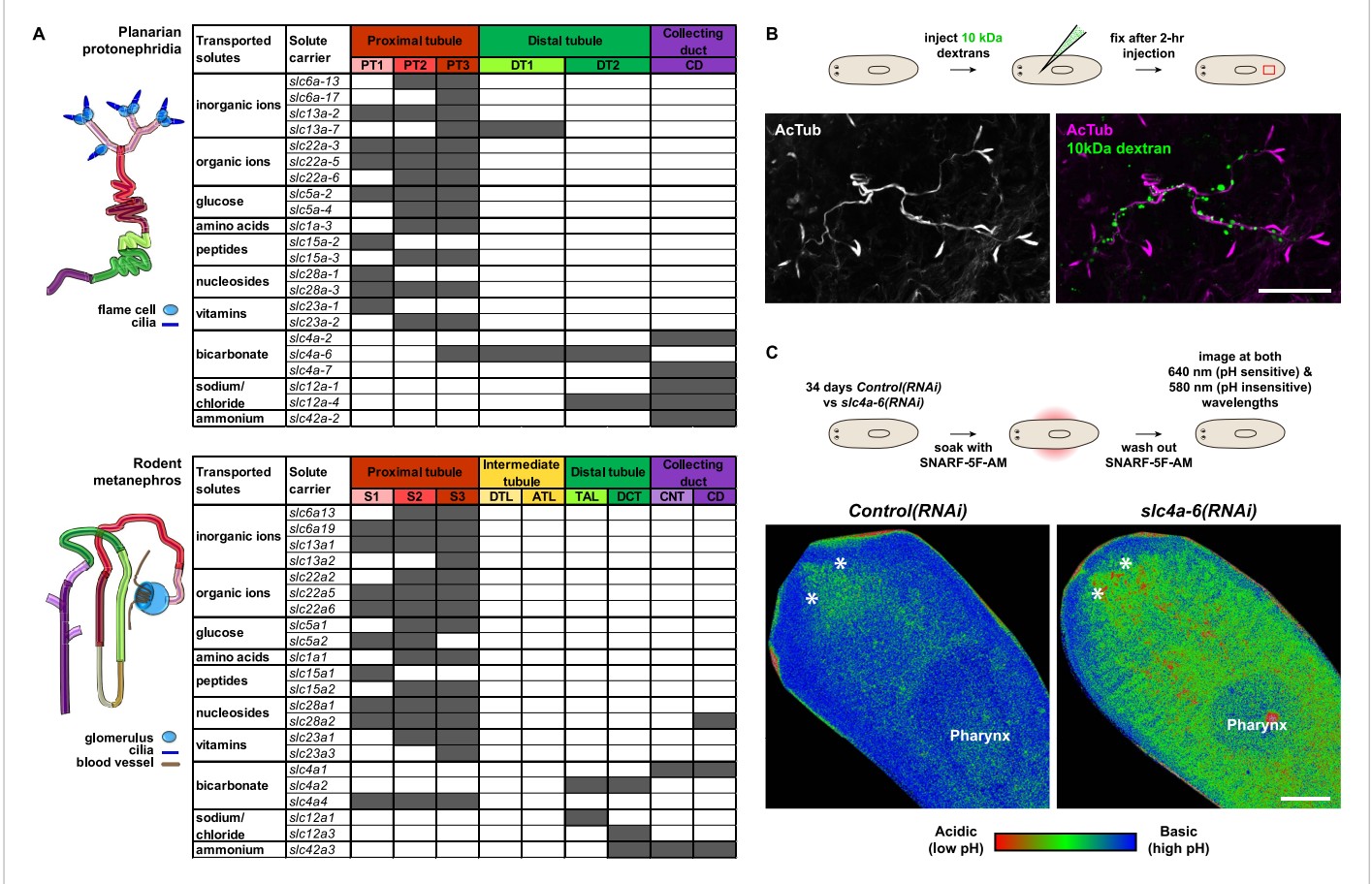

**Figure 3**. Extensive structural and functional homology between planarian protonephridia and vertebrate nephrons. (**A**) Tables summarize expression domains of selected *slc* genes in planarian protonephridia and rodent metanephros. Cartoons showing segmental organization of planarian protonephridia and rodent metanephros are on the left. Gray color in the tables indicates expression domain of *slc* in planarian protonephridia and rodent metanephros. Planarian *slc* sequence nomenclature (e.g., *slc1a-3*) doesn't reflect direct orthology to the mammalian counterparts. Abbreviations for segments of protonephridia are as follows: PT1, PT2, and PT3, segments of the proximal tubule (PT); DT1 and DT2, segments of the DT; CD, the collecting duct. Abbreviations for segments of the metanephros are as follows: S1, S2, and S3, segments of the PT; DTL, descending thin limb; ATL, ascending thin limb; TAL, thick ascending limb; DCT, distal convoluted tubules; CNT, connecting tubule; CD, collecting duct. (**B**) Fluorescent overlay of reabsorbed dextran with PT marker (AcTub). (**C**) pH$_i$ reporter assay using SNARF-5F-AM in *Control(RNAi)* and *slc4a-6(RNAi)*.

are the CKDs. We assembled a small library of putative planarian orthologues of human CKD genes (**Supplementary file 4**). This list included the nephrocystins, causative genes of nephronophthisis (NPHP), one of the most frequent genetic causes of chronic renal failure in children and young adults (**Hildebrandt and Otto, 2000**; **Salomon et al., 2009**). The *S. mediterranea* genome harbors homologs to human NPHP1-9, except for NPHP2 and NPHP3 (**Figure 5—figure supplement 1**). RNAi-screening of the library revealed strong edema formation in *Smed-NPHP5*, *Smed-NPHP6*, and *Smed-NPHP8* knockdown animals (**Figure 4A**), suggesting a protonephridial function for these genes. Consistently, we detected severe structural alterations of protonephridial tubules in *NPHP(RNAi)* animals, particularly of the proximal segment. RNAi animals presented striking clump-like accumulations of PT marker expressing cells (**Figure 5A,B**, **Videos 4**, **5**) instead of the fine terminal ramifications of PTs in controls. High-resolution imaging confirmed the presence of abnormally high numbers of densely packed PT cells (**Figure 5—figure supplement 2**). The protonephridial lumen was severely disorganized within such aggregates (**Figure 5C**). Instead of strong and continuous luminal labeling throughout the coiled PT segments of controls, labeling was weak and fragmented. The weak single-line labeling outside of aggregates (**Figure 5C**) and the much weaker cilia staining (AcTub) in *NPHP(RNAi)* animals (**Figure 5A**) suggested general lumen defects. EM images revealed frequent

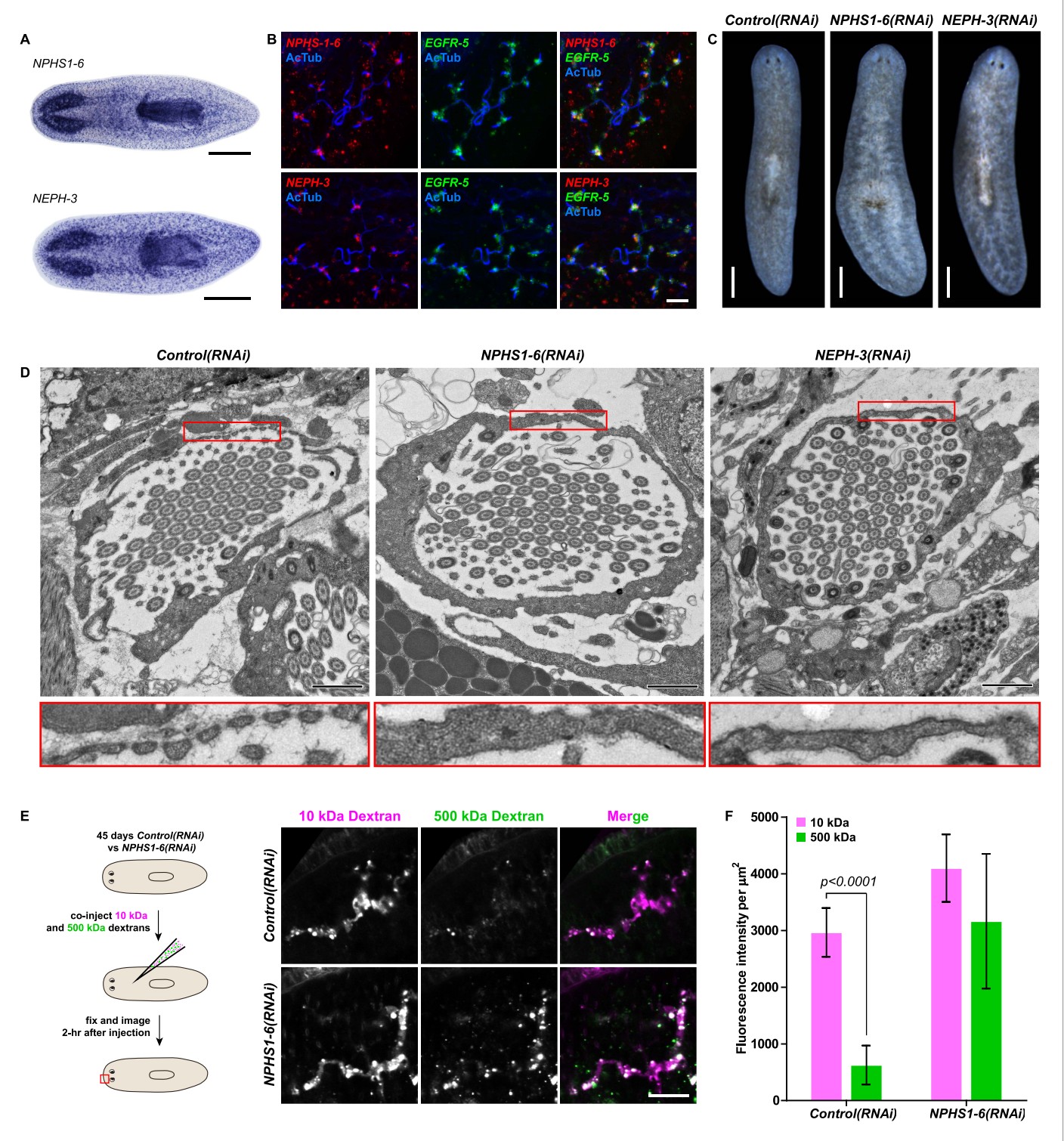

**Figure 4**. Vertebrate slit-diaphragm components are expressed in planarian flame cells and are required for the maintenance of their filtration diaphragm. (**A**) Whole-mount expression patterns of indicated marker genes by in situ hybridization (NBT/BCIP development). Scale bars: 500 μm. (**B**) Fluorescent overlay of indicated gene (red) with flame cell marker *EGFR-5* and AcTub staining. Images are maximum projections of confocal Z-sections. Scale bars: 50 μm. (**C**) Live images show edema in intact *NPHS1-6(RNAi)* and *NEPH-3(RNAi)* animals. Scale bars: 500 μm. (**D**) TEM images show cross-section through a flame cell in intact *Control(RNAi)*, *NPHS1-6(RNAi)* and *NEPH-3(RNAi)* animals. Inset shows a high magnification of the filtration diaphragm. Scale bar:
*Figure 4. continued on next page*

*Figure 4. Continued*

1 μm. (**E**, **F**) Ultrafiltration assay assesses the ultrafiltration capacity in *NPHS1-6(RNAi)* animals. (**E**) Representative images show dextran uptake in the animals that co-injected with 10 kDa and 500 kDa fluorescently labeled dextran. Scale bar: 50 μm. (**F**) Quantification of small and large dextran uptake.

The following figure supplements are available for figure 4:

**Figure supplement 1**. Slit-diaphragm components in the planarian *S. mediterranea*.

**Figure supplement 2**. *NPHS1-6* is not required for flame cell viability during normal homeostasis, as well as regeneration.

**Figure supplement 3**. *NPHS1-6* is required for de novo formation of filtration diaphragm during regeneneration.

basal body mislocalizations to non-luminal membrane domains and cell intrusions into the lumen, which both indicate a loss of normal tubular cell polarity (*Figure 5—figure supplement 3*). Altogether, the accumulation of morphologically abnormal tubule cells and concomitant loss of luminal connectivity present striking morphological parallels to the *NPHP* loss-of-function phenotype in humans, suggesting that planarian protonephridia can develop cyst-like structures.

## Protonephridial cysts originate from direct proliferation of protonephridial progenitors

Sustained cell proliferation in the renal tubules is a hallmark of cystic kidneys in humans and the severity of the phenotype correlates with the ectopic proliferation level (*Wilson and Goilav, 2007*). We used BrdU pulse labeling to determine whether cell proliferation was involved in the formation of the observed cyst-like structures (*Figure 6A*). In controls, we found occasional cells double positive for BrdU and the protonephridial progenitor marker *Smed-POU2/3* (*Scimone et al., 2011*) in the vicinity of tubules (*Figure 6A*), consistent with the emerging view that all planarian cell types derive from the proliferation of specific progenitor classes within the neoblast population (*Cowles et al., 2013*; *Adler et al., 2014*; *Scimone et al., 2014*; *van Wolfswinkel et al., 2014*). In *NPHP8(RNAi)* animals, the number of BrdU/*POU2/3* double-positive cells in the vicinity of cell accumulations was notably increased (*Figure 6A*), and in situ analyses further confirmed the progressive accumulation of protonephridial progenitors (*Figure 6—figure supplement 1A–D*). To probe the magnitude of

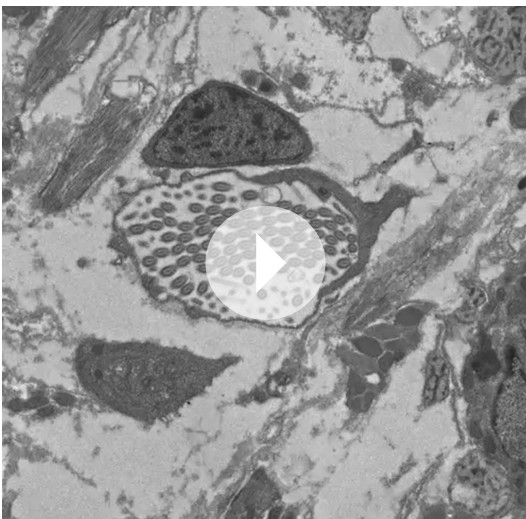

**Video 2.** Flame cell morphology in *Control(RNAi)* animal. Serial TEM images showing flame cell in *Control (RNAi)* animal.

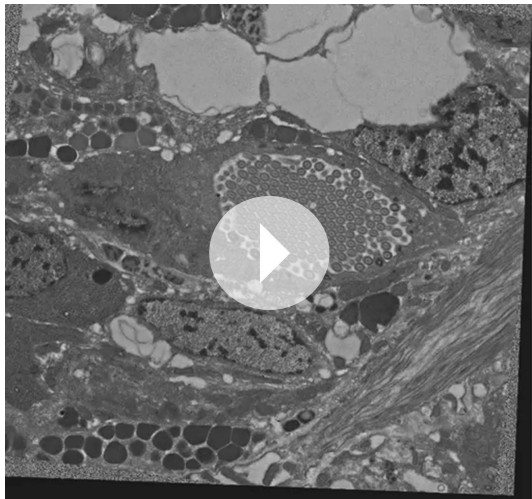

**Video 3.** Flame cell morphology in *NPHS1-6(RNAi)* animal. Serial TEM images showing flame cell in *NPHS1-6(RNAi)* animal.

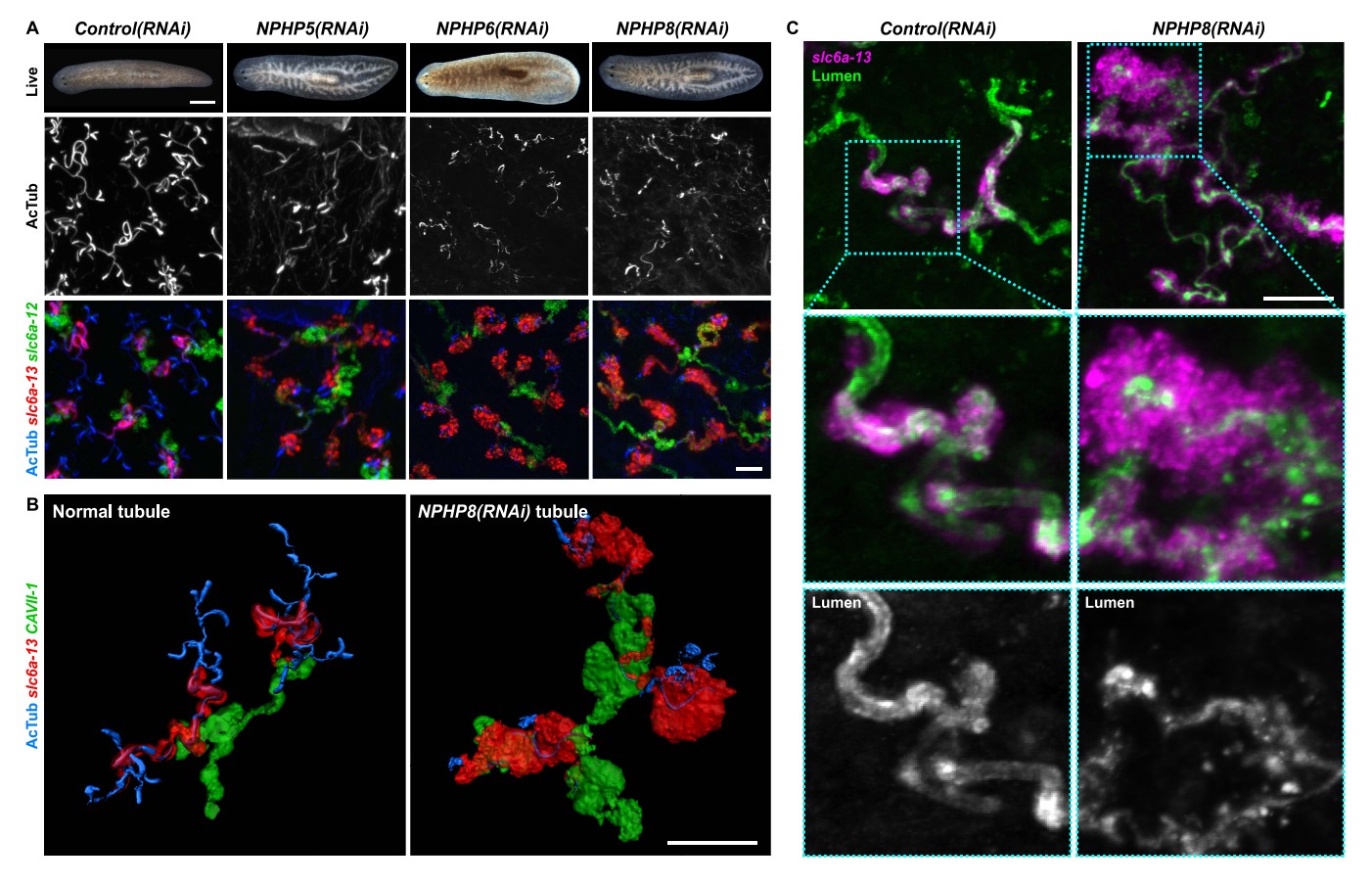

**Figure 5**. Down-regulation of nephrocystin members leads to the formation of cyst-like structure in protonephridia. (**A**) Protonephridial defects in *NPHP5 (RNAi)*, *NPHP6(RNAi)* and *NPHP8(RNAi)* animals. Top panel: live images show edema in intact RNAi animals. Scale bars: 500 μm; middle panel: monochrome showing AcTub staining; bottom panel: fluorescent overlay of AcTub staining with PT2 and PT3 marker (*slc6a-13*) and DT marker (*slc6a-12*). Scale bars: 50 μm. (**B**) 3D rendering images showing normal tubule and cystic-like tubule in *Control(RNAi)* and *NPHP8(RNAi)* animals, respectively. 3D rendering was performed in IMARIS. Scale bars: 50 μm. (**C**) Fragmented lumen in enlarged protonephridial tubule. Fluorescent overlay of PT2 and PT3 marker *slc6a-13* and lumen marker (a customized rabbit antiserum recognized unknown epitope) in intact *Control(RNAi)* and *NPHP8(RNAi)* animals. Scale bars: 50 μm. Images in (**A**) and (**C**) are maximum projections of confocal Z-sections.

The following figure supplements are available for figure 5:

**Figure supplement 1**. Nephrocystins in the planarian *S. mediterranea*.

**Figure supplement 2**. Abnormal tubular enlargement in *NPHP8(RNAi)* animals.

**Figure supplement 3**. Ultrastructure of the PT in *NPHP(RNAi)* animals.

the overproliferation effect, we carried out whole-mount staining with the G2/M-phase marker phospho-Histone H3 (H3P) and found a global increase in cell proliferation in *NPHP(RNAi)* animals (*Figure 6B*). To ask whether these effects were specific to protonephridial progenitors or globally affected all progenitor classes, we quantified the relative fraction of proliferation in protonephridial- ($POU2/3^+$/$smedwi$-$1^+$/$H3P^+$), neuronal- ($pax6A^+$/$smedwi$-$1^+$/$H3P^+$) (*Wenemoser et al., 2012*; *Scimone et al., 2014*) and intestinal ($HNF4^+$/$smedwi$-$1^+$/$H3P^+$) (*Wagner et al., 2011*; *Scimone et al., 2014*) progenitor classes (*Figure 6—figure supplement 1C,D*, *Figure 6—figure supplement 2*). Whereas the fraction of proliferating protonephridial progenitors was increased in both *NPHP5(RNAi)* and *NPHP8(RNAi)* animals, we found no change in the fraction of proliferating neuronal progenitors and even a slight decrease in intestinal progenitor proliferation (*Figure 6C–E*). The observation that all

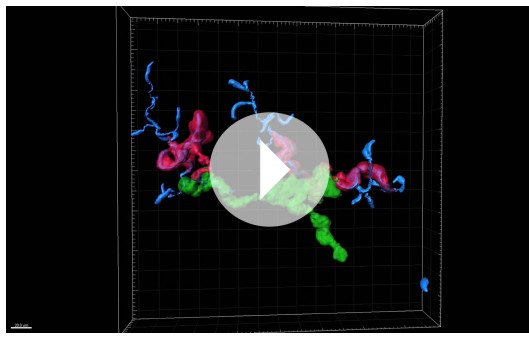

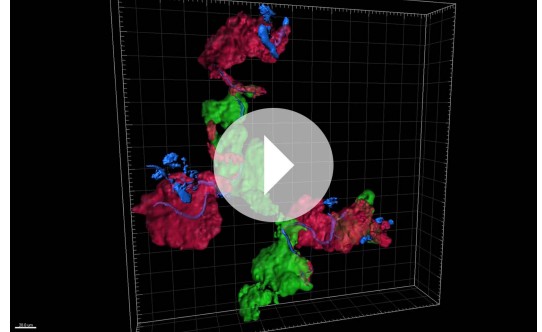

**Video 4.** 3D rendering of normal protonephridial tubule in *Control(RNAi)* animal.

**Video 5.** 3D rendering of cystic-like protonephridial tubule in *NPHP8(RNAi)* animal.

cases of ectopic BrdU-incorporation in the normally division-devoid area anterior to the photo-receptors were limited to *POU2/3*[+] protonephridial progenitors (*Figure 6—figure supplement 1E*) further supports the protonephridial specificity of the overproliferation response. Altogether, these results demonstrate that loss of function of planarian *NPHP* genes selectively increased the proliferation of protonephridial progenitors.

To test whether the level of proliferation determined the severity of the phenotype as observed in humans, we made use of the facile manipulation of global cell proliferation levels in the planarian system (*Figure 6F*, *Figure 6—figure supplement 3A*). Lethally or sub-lethally irradiated animals were used to examine the effects of abolished or reduced proliferation, respectively (*Wagner et al., 2012*), while animals on an increased feeding regiment provided an opportunity to examine the effects of above-baseline proliferation (*Kang and Sanchez Alvarado, 2009*). We found that edema development in *NPHP8(RNAi)* animals was faster and more severe under the increased proliferation condition, yet significantly diminished or even abolished under reduced or no proliferation, respectively (*Figure 6G–J*, left). The quantification of the confocally projected area of protonephridial marker expression domains (*slc6a-13* and *CAVII-1*) as a direct cell accumulation metric (*Figure 6G–J*, right; *Figure 6—figure supplement 3B*) also showed equal dependency on proliferation rates, thus indicating that the development of planarian NPHP phenotypes is tightly associated with cell proliferation.

In light of the striking morphological and ontological parallels between protonephridial and human *NPHP* loss-of-function phenotypes, we now refer to the observed structural alterations in planarian protonephridia as cysts.

## Cilia-driven fluid flow is required for tubular cell homeostasis in planarian protonephridia

Cilia as flow sensors play a critical role in the ontogeny of human CKDs (*Hildebrandt and Otto, 2005*; *Hildebrandt and Zhou, 2007*; *Kotsis et al., 2013*). *NPHP(RNAi)* planarians display severe defects in cilia-driven gliding motility (*Figure 7A,B*), prompting us to investigate a possible involvement of cilia in the ontogeny of planarian protonephridial defects. Direct visualization of axonemes in *NPHP(RNAi)* animals indeed confirmed structural cilia defects, which appeared shorter (*NPHP5(RNAi)*) or much reduced in density (*NPHP6/8(RNAi)*) (*Figure 7C*). EM images revealed abnormal localization of centrioles as well as axonemal abnormalities in ciliated cells under *NPHP5/6/8(RNAi)* (*Figure 5—figure supplement 3*). Together with the broad resemblance between *NPHP5/6/8* expression patterns and typical cilia genes (*Rink et al., 2009*; *Glazer et al., 2010*) (*Figure 5—figure supplement 1*), these data conclusively demonstrate that knockdown of planarian NPHP-genes causes not only protonephridial cyst formation, but also structural defects in cilia.

Therefore, we decided to systematically test possible mechanistic roles of cilia in planarian cyst ontogeny. If cilia were generally required for maintaining the structure/function of protonephridia, then all disruptions of cilia structure should cause cystic phenotypes. We therefore knocked down *Smed-IFT88*, a component of the intraflagellar transport machinery. As previously shown

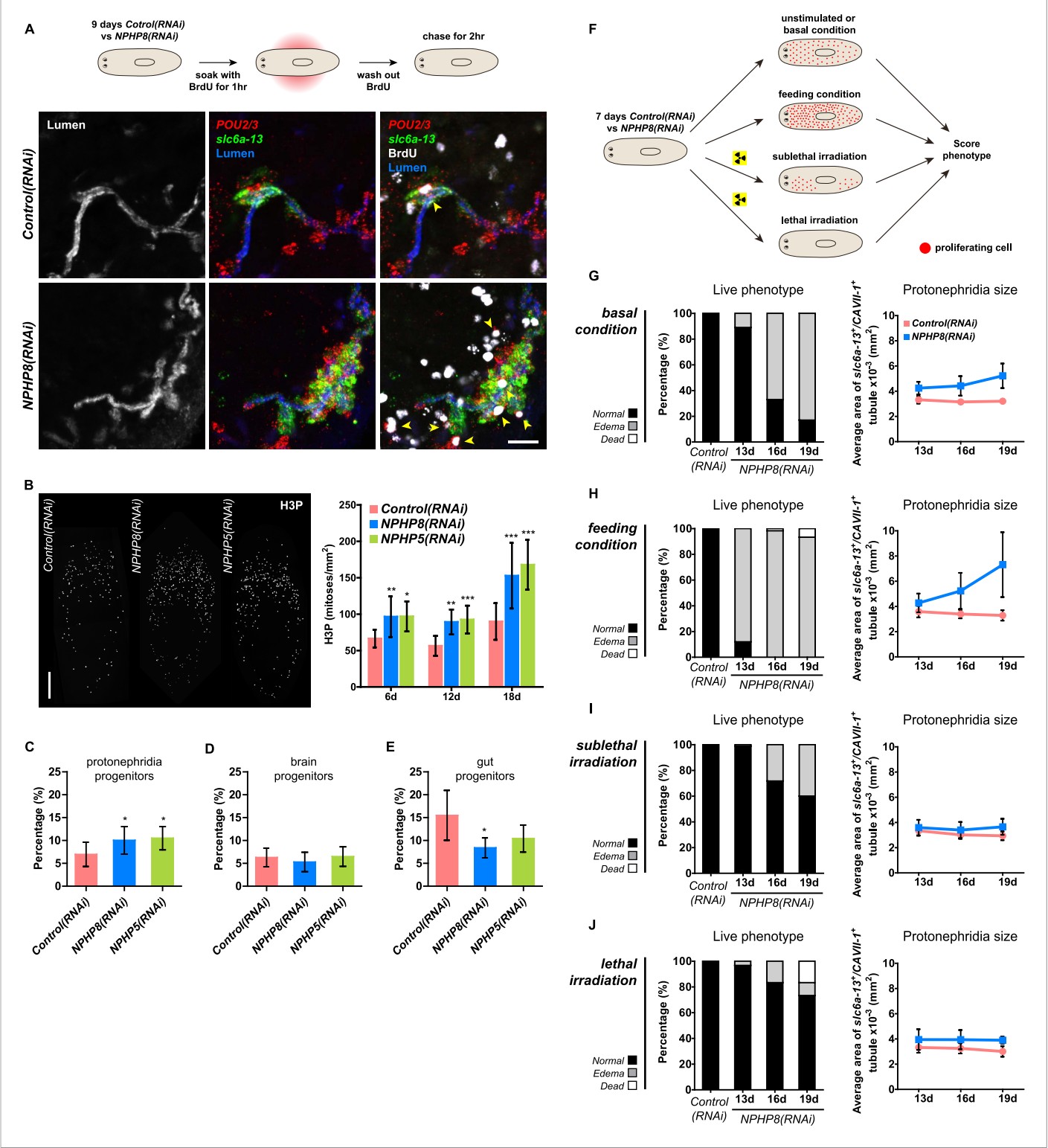

**Figure 6.** Cystogenesis in planarian protonephridia results from direct proliferation of protonephridia progenitors and requires the presence of stem cells. (**A**) BrdU pulse-chase experiment shows the presence of diving protonephridial progenitors in the proximity of protonephridial tubule in *Control(RNAi)* and *NPHP8(RNAi)* animals. Yellow arrowhead indicates *POU2/3+*/BrdU+ cell. Scale bars: 25 μm. (**B**) Increased global proliferation in *NPHP5(RNAi)* and *NPHP8(RNAi)* animals is displayed by immunostaining of mitotic marker phospho-Histone H3 (H3P). Scale bars: 500 μm. * p < 0.05; ** p < 0.01; *** p < 0.001 vs control. The time points in the bar graph indicate the number of day after the last dsRNA introduction. (**C–E**) Quantification of (**C**) dividing protonephridial progenitors (*POU2/3+*/H3P+), (**D**) diving neuronal progenitors (*pax6A+*/H3P+) and (**E**) diving gut progenitors (*HNF4+*/H3P+) among diving

*Figure 6. continued on next page*

*Figure 6. Continued*

cells (H3P⁺) in indicated RNAi animals at 18 day after last RNAi introduction. * p < 0.05 vs control. (**F–J**) Effect of proliferation and the requirement of neoblasts on cyst formation in planarian protonephridia. (**F**) Schematics demonstrates experimental strategy for panel **H–J**. 7 day post RNAi feeding animals were either fed with liver to induce cell proliferation or subjected to sublethal or lethal doses of irradiation to reduce or eliminate neoblasts. Scoring live phenotype as well as measuring the average size of each protonephridial unit was used to evaluate the severity of cystic phenotype. Temporal succession of indicated phenotypes (left) and quantification of average area of each *slc6a13*⁺/*CAVII-1*⁺ tubule (right) in *Control(RNAi)* and *NPHP8(RNAi)* animals under (**G**) basal condition (only RNAi feeding), (**H**) basal condition plus extra feeding with liver, (**I**) basal condition plus sub-lethal irradiation to reduce the number of neoblasts, and (**J**) basal condition plus lethal irradiation to completely eliminate neoblasts. The time points in the bar graphs (**G–J**) indicate the number of the day after the first dsRNA introduction.

The following figure supplements are available for figure 6:

**Figure supplement 1**. Increase of protonephridial progenitors during cystogenesis in planarian protonephridia.

**Figure supplement 2**. Gut and brain progenitors in NPHP8(RNAi) animals.

**Figure supplement 3**. The severity of cystic phenotype in protonephridia depends on proliferation rate and requires the presence of stem cells.

(*Rink et al., 2009*), *IFT88(RNAi)* animals lost their cilia-dependent gliding ability (*Figure 7—figure supplement 1A*), developed massive tissue edema and had severely shortened cilia (*Figure 7E,E′*). Interestingly, *IFT88(RNAi)* animals also developed cystic protonephridia (*Figure 7E″*) and cystogenesis in *IFT88(RNAi)* animals was also associated with increased proliferation (*Figure 7—figure supplement 2*) and the abnormal accumulation of protonephridial progenitors (*Figure 7E‴*). These results therefore demonstrate that disruption of cilia is sufficient for cyst development in planarians.

In contrast to adult mammalian kidneys that contain only immotile sensory cilia (*Webber and Lee, 1975*), the excretory systems of planarians and many other lower vertebrates possess motile cilia (*Figure 7—figure supplement 1B*; [*McKanna, 1968a, 1968b*; *Ishii, 1980a, 1980b*; *Lacy et al., 1989*; *Kramer-Zucker et al., 2005*; *Rink, 2013*]) that collectively drive fluid flow into and through the tubules (*White, 1929*; *Pontin, 1964*; *Warner, 1969*; *Kramer-Zucker et al., 2005*). Hence cilia-dependent cyst formation might reflect either a requirement of cilia as flow generators and/or as flow sensors. We first sought to disrupt ciliary beating without gross changes in cilia length or structure. We therefore targeted two planarian homologues of Primary Ciliary Dyskinesia (PCD) disease genes, a rare ciliopathy causing general cilia immobility in humans (*Badano et al., 2006*) (*Figure 7—figure supplement 1B,C*). Disrupting the function of *Smed-DNAHβ-1* and *Smed-LRRC50* by RNAi led to abnormal gliding ability (*Figure 7—figure supplement 1A*) due to loss of ciliary beating (*Figure 7H–I*; *Videos 6–8*), while cilia length and structure appeared unaffected (*Figure 7F′–G′*). Interestingly, *DNAHβ-1-* and *LRRC50(RNAi)* animals also developed edema and formed protonephridial cysts (*Figure 7F,G,F″,G″,F‴,G‴*). These results indicate that reduced ciliary beating rate without change in cilia structure is sufficient to cause the cystic phenotype in planarian protonephridia. Together, these results suggest that cilia-driven fluid flow is crucial to orchestrate tubular cell homeostasis in planarian protonephridia.

## Discussion

Our study has uncovered remarkable molecular and functional similarities between the basic functions of planarian protonephridia and those of the vertebrate nephrons. First, key constituents of the vertebrate podocyte slit diaphragm (*NPHS1* and *NEPH1*) are not only expressed, but also required for planarian flame cell ultrafiltration function (*Figure 4*). Our data, combined with recent studies in flies (*Weavers et al., 2009*; *Zhuang et al., 2009*) demonstrate a deep evolutionary conservation of *NPHS1* and *NEPH1* function in ultrafiltration barrier maintenance in planarian flame cells, insect nephrocytes, and vertebrate podocytes. Second, the systematic gene expression mapping of *slc* families in the planarian protonephridial tubule revealed striking topological parallels to the vertebrate pronephros/metanephros (*Figure 3A*). Our results demonstrate functional homologies between sub-segments of the vertebrate nephrons and planarian protonephridia. Third, the extensive structure/function homologies between planarian protonephridia and vertebrate nephrons encompass common pathologies. We found that many human kidney disease genes cause similar pathologies in planarian

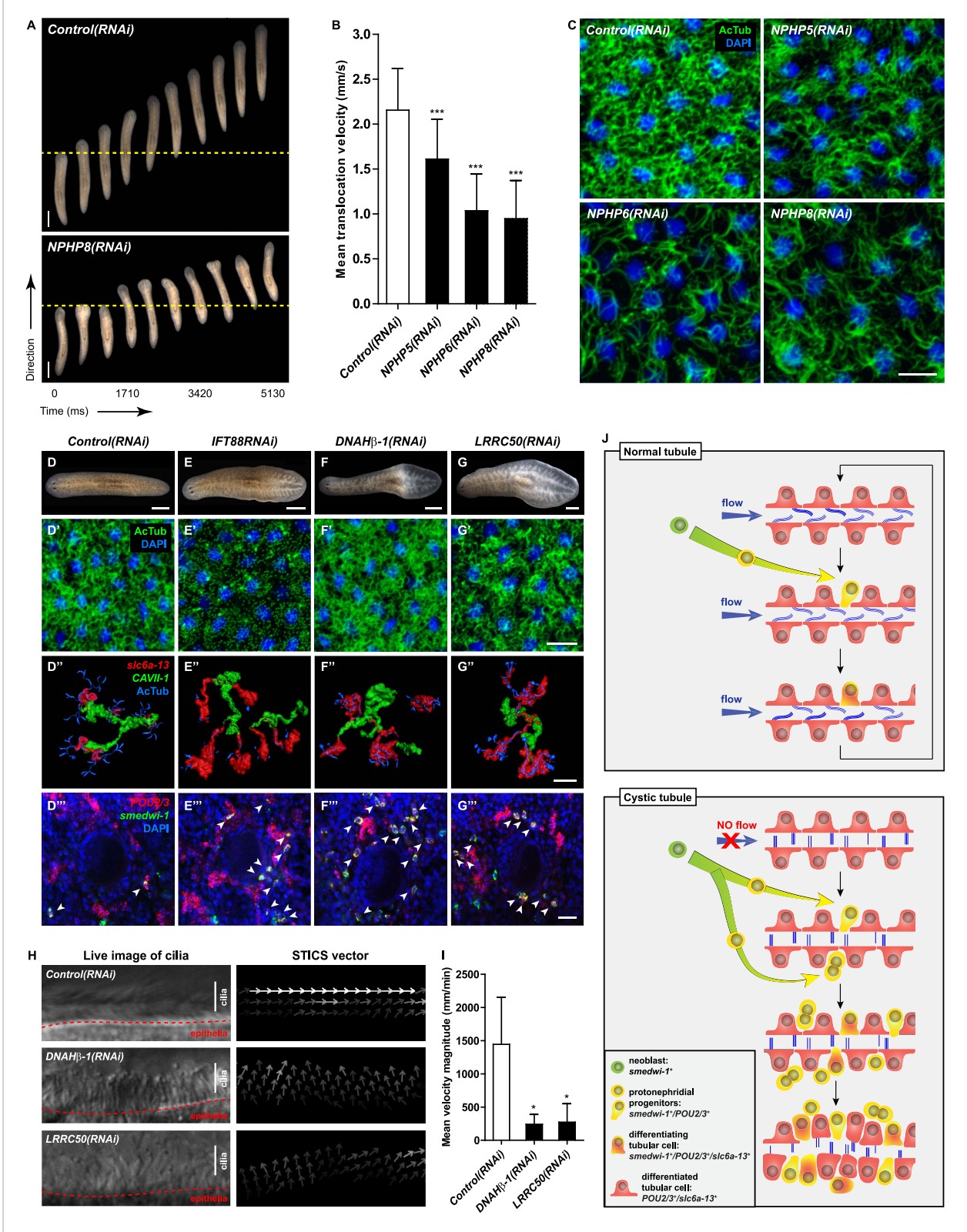

**Figure 7**. Cystic phenotype in protonephridia is cilia-and fluid flow-dependent. (**A**) Series of live images show gliding mobility in *Control(RNAi)* and *NPHP8(RNAi)* animals. Yellow dot line provides a spatial reference to illustrate progress of animal. Scale bar: 1 mm. (**B**) Quantification of translocation speed in indicated RNAi animals. Error bar, SD; *** p < 0.001 vs control. (**C**) Fluorescent overlay of ventral cilia (AcTub) with nucleus marker (DAPI) in *Figure 7. continued on next page*

*Figure 7. Continued*

indicated RNAi animals. Scale bar: 10 μm. (**D–G**) Live images show bloating phenotype in *IFT88(RNAi)*, *DNAHβ-1(RNAi)*, and *LRRC50(RNAi)* animals. Scale bar: 500 μm. (**D'–G'**) Fluorescent overlay of ventral cilia (AcTub) with nucleus marker (DAPI) in *IFT88(RNAi)*, *DNAHβ-1(RNAi)*, and *LRRC50(RNAi)* animals. Scale bar: 10 μm. (**D''–G''**) 3D rendering showing fluorescent overlay of AcTub staining with PT2 and PT3 marker (*slc6a-13*) and DT marker (*CAVII-1*) in *Control(RNAi)*, *IFT88(RNAi)*, *DNAHβ-1(RNAi)*, and *LRRC50(RNAi)* animals. Scale bar: 50 μm. (**D'''–G'''**) Magnified view shows fluorescent overlay of *POU2/3* with pan stem cell marker (*smedwi-1*) in the region surrounding photoreceptor. White arrowhead shows *POU2/3*[+]/*smedwi-1*[+] cell. Scale bar: 25 μm. (**H–I**) Abnormal cilia beating in *DNAHβ-1(RNAi)*, and *LRRC50(RNAi)* animals. (**H**) Left panel: live images show cilia beating along the lateral body edge of the planarian head region; Right panel: vector map generated by Spatiotemporal image correlation spectroscopy (STICS) analysis shows velocity magnitude and beating pattern of cilia. The brightness of the vector represents the velocity magnitude of the cilia: brighter vector, stronger ciliary beating or vice versa. (**I**) Quantification of ciliary velocity magnitude in indicated RNAi animals. * $p < 0.05$ vs control. (**J**) Cartoon represents working model of cyst formation in the planarian protonephridia. In normal tubule, protonephridial tubular cell turnover is maintained by integration of protonephridial progenitors, originated from the neoblasts, into the tubule. During this process, cilia-driven fluid flow is required for the maintenance of tubular geometry. Obstruction of fluid flow by disrupting cilia function leads to protonephridial cystogenesis that characterized by abnormal proliferation of protonephridial progenitors, tubular enlargement and disorganization.

The following figure supplements are available for figure 7:

**Figure supplement 1**. Primary Ciliary Dyskinesia (PCD) genes in the planarian *S. mediterranea*.

**Figure supplement 2**. Cystogenesis in planarian protonephridia under *IFT88(RNAi)* is correlated with increased proliferation.

protonephridia, most notably cyst formation upon *NPHP* knockdown. Altogether, our comprehensive molecular, structural, and functional characterization of the planarian protonephridia strongly supports a common evolutionary origin for the excretory systems of bilateral metazoans.

## Planarian protonephridia represent a novel invertebrate model for human kidney diseases

An important consequence of these findings is that planarian protonephridia provide an invertebrate model for human kidney diseases. Both human nephrons and planarian protonephridia rely on pressure-driven ultrafiltration with subsequent filtrate modification during its passage through the nephridial tubules. This common design principle is significant for two reasons: first, many human kidney pathologies involve filtrate flow (see below); and second, the excretory systems of existing invertebrate model systems do not recapitulate this critical aspect. In fruitflies and other insects, ultrafiltration and filtrate modifications are carried out in separate cell types and organs (ultrafiltration: nephrocytes; filtrate modification: Malpighian tubules) (*Dow and Romero, 2010*). *C. elegans* has a highly derived excretory system consisting of just a single cell (*Buechner, 2002*). Both are therefore unsuitable for modeling flow-related kidney diseases, including polycystic kidney disease (PKD). PKD is one of the most common, life-threatening genetic diseases that affects millions worldwide

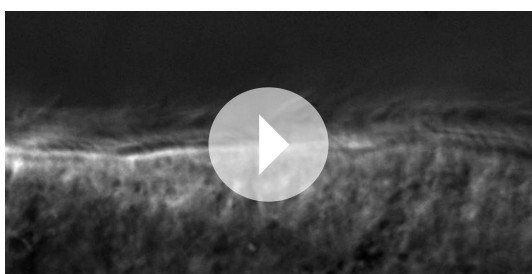

**Video 6.** Cilia beating in *Control(RNAi)* animal. Serial images show beating of the cilia along the lateral body edge of the planarian head region in *Control(RNAi)* animal.

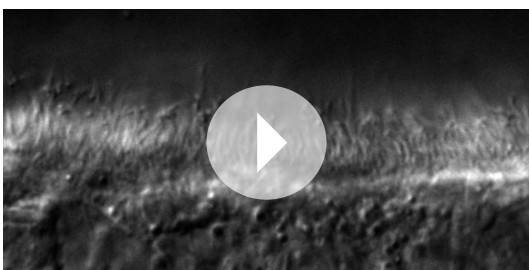

**Video 7.** Cilia beating in *DNAHβ-1(RNAi)* animal. Serial images show beating of the cilia along the lateral body edge of the planarian head region in *DNAHβ-1(RNAi)* animal.

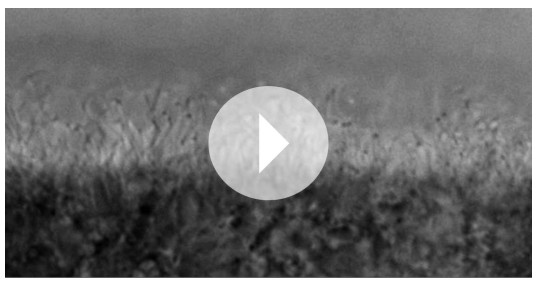

**Video 8.** Cilia beating in *LRRC50(RNAi)* animal. Serial images show beating of the cilia along the lateral body edge of the planarian head region in *LRRC50(RNAi)* animal.

(*Wilson, 2004a, 2004b, 2011*). The pathology of PKD is characterized by the massive overproliferation of kidney cells, coupled with aberrant differentiation and the formation of fluid-filled cysts (*Wilson, 2004a, 2004b, 2011*). In contrast to *C. elegans* and *D. melanogaster*, our results demonstrate that ultrafiltration and filtrate flow are as important in maintaining excretory organ homeostasis in both planarians and humans.

## Cilia regulate excretory organ homeostasis in both planarians and vertebrates

Such parallels between human nephrons and planarian protonephridia may seem surprising at first, given that the two excretory systems are phenotypically very different. In vertebrates, individual nephron units combine within the kidney as a single organ, whereas in planarians, protonephridial units are pervasively present throughout the body (*Rink et al., 2011*). Further, in vertebrates, it is blood pressure and, ultimately, the heart that drives primary filtrate across the glomerular basement membrane and into the nephron. Planarians lack a circulatory system and therefore cannot solely rely on pressure-driven flow generation. The dense and vigorously beating cilia bundles of flame cells, and the multiciliated lining of the PT suggest instead that motile cilia are responsible for driving protonephridial ultrafiltration and flow. In fact, previous studies of protonephridia in other invertebrates strongly support this conclusion, including the direct observation of fluid flow cessation upon pharmacological inhibition of ciliary beating in the rotifer *Asplanchna* (*Pontin, 1964*; *Warner, 1969*).

The nephrons of the adult human kidney also bear cilia. However, in contrast to the multiciliated and flow generating cilia of the planarian PT, the cells of the adult mammalian nephron bear a single, immotile primary cilium protruding into the lumen (*Webber and Lee, 1975*). Despite these structural and functional differences, we find that cilia are similarly crucial for the maintenance of organ homeostasis in both systems. In humans, loss of function of many genes involved in cilia biogenesis/function (*Eley et al., 2005*; *Hildebrandt and Otto, 2005*; *Winyard and Jenkins, 2011*; *Kotsis et al., 2013*) leads to overproliferation and cyst formation, including for example the NPHP proteins, that anchor the basal body to the cell membrane and are required for primary cilia formation (*Betleja and Cole, 2010*; *Craige et al., 2010*; *Omran, 2010*; *Williams et al., 2011*). The importance of cilia is thought to be due to their function as flow sensors (*Winyard and Jenkins, 2011*; *Kotsis et al., 2013*). Flow-induced bending is hypothesized to cause calcium influx via stretch sensitive polycystin channels, which subsequently inhibits cell proliferation via still unknown mechanisms (*Praetorius and Spring, 2001, 2003*; *Nauli et al., 2003*; *Praetorius et al., 2004*). Unexpectedly, in spite of the very different structure and function of planarian protonephridial cilia, we found that they are similarly critical in maintaining form and function of the organ system. First, knockdown of the planarian homologues of NPHP proteins lead to overproliferation of protonephridial progenitors. Second, the ablation of cilia by *IFT88(RNAi)* or the inhibition of ciliary beating by interfering with axonemal dyneins leads to protonephridial progenitor overproliferation. Moreover, in both cases, accumulating protonephridial progenitors form dense aggregates with disorganized lumens that quickly compromise the osmoregulatory functions of the organs, leading to edema formation. Hence the disruption of planarian cilia leads to phenotypically very similar alterations as PKD in vertebrates. We therefore conclude that cilia-mediated flow sensing constitutes an ancient mechanism for maintaining excretory organ homeostasis.

## Cilia play a dual role as flow generator and flow sensor in the planarian protonephridia

Planarians have lost central components of centrosome duplication, which should therefore preclude their ability to form primary cilia (*Azimzadeh et al., 2012*). Since planarians only appear to harbor motile cilia, the corollary of this argument would be that the cilia of the protonephridial tubule have a dual role as flow generators and as flow sensors. Our data cannot conclusively address this point, because the

currently available experimental paradigms do not allow the requisite uncoupling between cilia and flow. NPHP proteins are generally thought to only be essential for immotile sensory cilia (*Eley et al., 2005*; *Badano et al., 2006*; *Ferkol and Leigh, 2012*) based on their involvement in anchoring basal body to the cell membrane (*Betleja and Cole, 2010*; *Craige et al., 2010*; *Omran, 2010*; *Williams et al., 2011*). Loss of NPHP 'sensory' machinery in the multiciliated tubule cells leads to cyst formation in planarians, indicating the putative flow-sensing role of protonephridial cilia. However, bronchiectasis, a classic phenotypic feature of PCD, has been found in many cases of NPHP (*Bagga et al., 1989*; *Bergmann et al., 2008*; *Moalem et al., 2013*; *Wolf, 2015*) as well as autosomal dominant PKDs (*Driscoll et al., 2008*; *Moua et al., 2014*), indicating impaired motor cilia function. Conversely, phenotypic features of defective sensory cilia, including retinitis pigmentosa and cystic kidneys, have been reported in PCD patients (*Kartagener and Horlacher, 1935*; *Saeki et al., 1984*; *Moore et al., 2006*). These observations suggest an under-appreciated overlap between NPHP gene functions in motile cilia with that of PCD genes in immotile cilia, which makes it difficult to segregate flow generating and flow sensing roles of protonephridial cilia. However, recent work in vertebrate systems indicates that the transduction of sensory cues from the environment is a universal characteristic of all cilia, including motile cilia (*Teilmann and Christensen, 2005*; *Shah et al., 2009*), suggesting the likely existence of flow sensing role of motile cilia in planarian protonephridia (*Schwartz et al., 1997*; *Teilmann and Christensen, 2005*; *Shah et al., 2009*; *Quarmby and Leroux, 2010*; *Takeda and Narita, 2012*).

In humans, polycystin-1 and polycystin-2 are key components of cilia-mediated flow sensing (*Praetorius and Spring, 2001*, *2003*; *Nauli et al., 2003*; *Praetorius et al., 2004*). As stretch-activated calcium channels, these proteins are thought to transduce the mechanical bending of the cilium into a chemical signal. Planarians have homologues of both polycystins and their knockdown leads to defects in ciliated dorsal mechanosensory neurons, but interestingly, not in protonephridia (Vu HTK and Rink JC, unpublished). Hence flow sensing in planarian protonephridia likely utilizes different mechanisms. The future elucidation of the underlying mechanisms may uncover general mechanistic links between flow sensing and flow generating in motile cilia.

## Cilia-driven fluid flow points to a shared ancestry of vertebrate and invertebrate excretory systems

Given the extensive evolutionary homology between excretory systems, how can the exclusive reliance on motile cilia driven flow generation in invertebrates be reconciled with their absence in adult vertebrate kidneys? The loss of motile cilia in vertebrate kidneys has been observed in coincidence with the increase in blood pressure in birds and mammals (*Marshall, 1934*). This raises the possibility that the evolution of an extensively developed circulatory system in mammals rendered the flow-generating role of motile cilia redundant, and subsequently led to their loss from the adult kidney. The evidence for this hypothesis can be found in the embryology of the vertebrate kidney, which, as Ernst Haeckel argued, may indeed hint at phylogeny (*Haeckel, 1866*). Many vertebrates display multiciliated flow generation during early developmental stages (*White, 1929*; *Lacy et al., 1989*; *Kramer-Zucker et al., 2005*). For instance, the pronephric tubules of zebrafish contain multiciliated epithelial cells, and interference with cilia obstructs flow (*Kramer-Zucker et al., 2005*). In humans, motile cilia may even play a role at early kidney development, since multiciliated tubular cells have been observed at fetal stages (*Zimmermann, 1971*; *Katz and Morgan, 1984*). Furthermore, a number of human pathologies such as crescentic and membranoproliferative glomerulonephritis (*Katz and Morgan, 1984*; *Hassan and Subramanyan, 1995*), as well as focal segmental glomerulosclerosis (*Katz and Morgan, 1984*) are associated with the appearance of multiciliated tubular cells. This suggests that the gene regulatory programs for the development of multiciliated cells remain intact and accessible even in vertebrates. Altogether, these observations suggest that cilia-driven pressure generation may be the ancestral state for excretory systems, and that the immotile primary cilia of the adult vertebrate kidney represents an evolutionary vestige of the cilia-driven flow filtration observed in more basal excretory organs.

## Structural/functional abnormalities of cilia lead to cyst formation in excretory organs of both planarians and vertebrates

Even though the exact flow sensing mechanisms remain to be identified, our data show that in planarians, these signals also affect cell proliferation. We observed a global increase of mitoses due to

the specific overproliferation of protonephridial progenitors whenever cilia or fluid flow were affected (*Figure 6B–E*). The aberrant proliferation and differentiation of these cells ultimately lead to dense cell accumulations with discontinuous lumens. In NPHP knockdown planarians, cysts primarily arise in the PT at early stage, but over time, progressively affect the DT as well (*Figure 6—figure supplement 2B*). The severity of the cystic phenotype intimately depends on global proliferation rates (*Figure 6F–J*), which, interestingly, is also the case in human CKDs (*Wilson and Goilav, 2007*). However, not all aspects of the planarian *NPHP(RNAi)* phenotypes phenocopy human NPHP. In human NPHP, cysts arise at the corticomedullary junction of the kidneys, meaning that they mostly develop from the distal convoluted and collecting tubules (*Hildebrandt and Zhou, 2007*). This discrepancy could be due to the differential distribution of cilia, which in planarians are only present in the PT (*Rink et al., 2011*). Accordingly, PT cysts could be the direct consequence of cilia dysfunction, while structural alterations of the DT could be a secondary consequence of overproliferation of protonephridial progenitors. Conversely, cilia are found on the apical surface of most epithelial cells lining the human nephrons with the exception of the intercalated cells interspersed along the CD (*Webber and Lee, 1975*). Nevertheless, why cysts predominantly affect the distal convoluted and collecting tubules only, but not other tubule parts in human NPHP is still unknown.

## Conclusion

Overall, the interplay between cilia and cell proliferation in protonephridial cyst formation is remarkably similar between humans and planarians. Therefore, in planarian protonephridia, cilia appear to generate a non-cell autonomous signal capable of regulating the proliferation of protonephridial progenitors and orchestrating their integration into the protonephridial tubules (*Figure 7J*). The search for the flow-regulated progenitor populations in the adult vertebrate kidneys is currently an active area of research (*Elger et al., 2003*; *Diep et al., 2011*; *McCampbell and Wingert, 2012*; *Rinkevich et al., 2014*). The mechanisms constituting the non-cell autonomous signal between tubule cells and progenitor cells is a second missing element in our current understanding of human CKDs and the homeostatic role of flow sensing in general. Given the striking evolutionary conservation of flow sensing and flow-dependent progenitor proliferation between planarians and vertebrates, it seems likely that also this elusive signal is conserved. In this regard, the high speed and low cost of high-throughput RNAi screening in planarians therefore provides a novel experimental paradigm for gene discovery and mechanistic studies of human kidney diseases.

## Materials and methods

### Planarian maintenance and irradiation

The CIW4 clonal line of *S. mediterranea* was maintained as described (*Cebria and Newmark, 2005*). 1 week starved animals were used for all experiments. For irradiation experiments, animals were exposed to 1250 or 6000 rads on a GammaCell 40 Exactor irradiator.

### Gene identification and cloning

Human, mouse, *Xenopus* and zebrafish protein sequences were used to find planarian homologs from *S. mediterranea* genome database via TBLASTN. Planarian homologs were then used for reciprocal BLAST against the human refseq to verify the homology. All genes were cloned from an 8 day regeneration time course cDNA library prepared as described previously (*Gurley et al., 2008*). Primers used for cloning are described in *Supplementary files 1, 4*.

### Phylogenetic analysis

The complete set of protein sequences were retrieved for human, mouse, and fly from Ensembl (release 76) (*Flicek et al., 2014*). The mosquito protein sequences were retrieved from Ensembl metazoa (release 23). Only the proteins corresponding to the longest isoform of each gene were considered for the analysis. The PFAM protein domains (PfamA-27.0) (*Finn et al., 2014*) were predicted for all those proteins from human, mouse, fly and mosquito and the planarian homologs of solute carriers using the InterProScan (version 5.4–47.0) tool (*Jones et al., 2014*). The solute carrier proteins were classified into their corresponding solute carrier family or clan groups based on the presence of the corresponding PFAM protein domain as described in the literature (*He et al., 2009*;

*Hoglund et al., 2011*). The predicted domain regions were extracted from those proteins and multiple sequence alignment was then performed for those extracted regions using clustalw2 (version 2.1, with default parameters) (*Larkin et al., 2007*). Using the sequence alignment, the bootstrapped neighbor joining trees (positions with gaps removed and corrected for multiple substitution) were constructed using clustalw2 (version 2.1) (*Larkin et al., 2007*).

## In situ hybridization and immunohistochemistry

Colorimetric and FISHs were performed as previously described (*Pearson et al., 2009*; *King and Newmark, 2013*). Following fluorescent or NBT/BCIP development, animals were incubated with anti-acetylated-Tubulin antibody (1:1000, Cell Signaling, Danvers, MA), anti-H3P (1:1000, Millipore), or a rabbit antiserum recognized unknown epitope to visualize the lumen of PT (1:500). Primary antibodies were detected with either Alexa-conjugated anti-rabbit antibodies (1:1000; Abcam) or HRP-conjugated anti-rabbit antibodies (1:1000; Jackson ImmunoResearch). NBT/BCIP developed whole-mount in situ specimens were mounted in mounting media containing 75% glycerol and 2 M urea. Fluorescent whole-mount in situ specimens were mounted in modified ScaleA2 containing 20% glycerol, 2.5% DABCO and 4 M urea (*Hama et al., 2011*). For cryosectioning, fluorescently stained whole-mounted animals were fixed overnight in 4% paraformaldehyde (in PBS) at 4°C, washed three times in PBS, equilibrated in 30% sucrose, frozen in OCT, and cryosectioned (10–20 μm).

## Imaging and image quantification

A Leica M205 Stereo Microscope was used for documenting live images, videos, and NBT/BCIP developed whole-mount in situ specimens. Zeiss LSM-510 VIS or LSM-700 Upright confocal microscopes were used to capture fluorescent whole-mount in situ specimens and image projections. To quantify the average size of each protonephridial unit and mitotic activity, individual worm was imaged and tiled on a Perkin Elmer Ultraview spinning disk microscope. Stitching and mitotic activity quantification was performed in FiJi using standard plugins (*Schindelin et al., 2012*). Worm area, protonephridial size and number were measured/counted using a custom signal to noise thresholding and seeded region grow plugins. Batching was performed using macros. Movement speed quantification was performed on video sequences (acquired at 17.5 Hz) using a custom thresholding plugin and Mtrack2 (*Klopfenstein and Vale, 2004*). For each tracked object, the initial position was subtracted from the final to determine an average translocation velocity. Average velocities were computed by weighting track averages by the length of the track. Plugins and macros are available at https://github.com/jouyun.

## BrdU labeling

BrdU was administered by soaking animals in 15 mg/ml BrdU and 3% DMSO (diluted in 0.1× Montjuic salts) for 1 hr as previously described (*Cowles et al., 2012*) and chasing for specified time. Animals were fixed and processed as in situ hybridization protocol except they were bleach in 6% $H_2O_2$ in PBSTx (0.5% Triton) for 3–4 hr under direct light. After in situ development, specimens were treated with 2 N HCl for 45 min at room temperature, and washed four times with PBSTx (0.3% Triton) for 1 hr. BrdU was detected using rat anti-BrdU antibody (1:1000; Abcam, Cat. No. ab6326). Primary antibody was detected with HRP-conjugated anti-rat antibody (1:1000; Jackson ImmunoResearch).

## Ultrafiltration and reabsorption assay

To assay ultrafiltration capacity of planarian protonephridia, 10 kDa tetramethylrhodamine-dextran (Molecular Probes, D-1817) and 500 kDa fluorescein-dextran (Molecular Probes, D-7136) at the concentration of 1 mg/ml were co-injected into the mesenchyme of the animals. After 2 hr, the animals were rinsed with an excess of 1× Montjuic salts, fixed in cold 4% paraformaldehyde (in 1× Montjuic salts), mounted in modified ScaleA2 and photographed using a Zeiss LSM-510 VIS confocal microscope. Dextran uptake was quantified by measuring the average fluorescence intensity per unit area using a standard signal to noise thresholding in Fiji (*Schindelin et al., 2012*). For immunostaining, after fixation, the samples were rinsed 3–4 times with PBSTx (0.3% Triton), incubated in blocking solution containing 5% horse serum in PBSTx (0.5% Triton) for 2 hr at room temperature, and then in anti-acetylated-Tubulin antibody (1:1000, Cell Signaling). Primary antibody was detected using Alexa-conjugated anti-rabbit antibodies (1:1000; Abcam).

## pH$_i$ reporter assay

Intracellular pH was measured using ratiometric pH dye SNARF-5F-AM (Molecular Probes, Cat. No. S-23923) at 5 μM (in DMSO with 20% wt/vol Pluronic F-127) as previously described (*Beane et al., 2011, 2013*). Animals were soaked for 1 hr, rinsed three times with an excess of 1× Montjuic salts, immobilized on the glass bottom dish using the microfluidic device and imaged at both 640 nm (pH sensitive) and 580 nm (pH insensitive) wavelengths using a LSM-700 Falcon confocal microscope. The ratio of 580/640 (used for controlling uneven dye uptake) was shown.

## High-speed video microscopy

To visualize cilia beating along the lateral body edge of the planarian head region, live worms are immobilized on the glass bottom dish using a microfluidic device and imaged on a Zeiss Axiovert 200 microscope under DIC optics using 63× objective. Series of images were captured at 250 frames per second with pixel number of 800 × 800 (exposure time is 3.97 ms) using an ORCA-Flash4.0 V2 C11440-22CU camera from Hamamatsu. Spatiotemporal image correlation spectroscopy was used to determine the speed of the cilia for each animal. In each time-lapse, 100 consecutive frames were manually selected in which the animal was stationary so that no image registration was required. A region of interest was manually drawn around the cilia in each time-lapse. The area outside this region was uniformly filled with the average intensity inside the region. Spatiotemporal correlation was then carried out in 32 × 32 pixel regions with a 16 pixel overlap between the regions to allow for highly localized motions to be accurately represented using the fast Fourier transform method. The average cilia displacement within the correlation image is represented by the maximum of the spatial cross-correlation between two images separated in time. The time correlation shift was a single frame, and all velocities were converted to micrometers per minute. This method was adapted from a previous paper (*Yi et al., 2011*), where it was implemented with custom plugins written in Java for ImageJ, available for download at (http://research.stowers.org/imagejplugins).

## Statistical analysis

Statistical analysis of the data was carried out in Excel. p values were determined using Student's *t*-test.

## EM

Specimens were prepared as following at 4°C on orbital rotator: (1) fix in cold 2.5% glutaraldehyde in 0.05 M or 0.1 M sodium cacodylate (contained 1 mM CaCl$_2$) for overnight; (2) wash in wash buffer (0.1 M sodium cacodylate buffer; 1 mM CaCl$_2$; and 1% sucrose) for 1 hr (3–4 exchanges); (3) fix in 1% Osmium tetroxide in 0.1 M sodium cacodylate buffer (+1 mM CaCl$_2$) for 2 hr; (4) wash in wash buffer for 1 hr (3–4 exchanges) and in distill water for 30 min (3–4 exchanges); (5) fix in 0.5% aqueous Uranyl Acetate (in dark) overnight; (6) wash in distill water for 30 min (3–4 exchanges); (7) and dehydrate in acetone 30% (20 min), 50% (20 min), 70% (overnight), 90% (20 min, 2 times), and 100% (20 min, 3 times). Specimens were then embedded in epon-araldite or Spurr's resin as follows: 25% resin/ acetone for 3 hr; 50% resin/acetone for 2.5 hr; 75% resin/acetone overnight; 100% resin without accelerator with microwave at 350 W for 3 min on/3 min off/3 min on for 1 day (2 exchanges); 100% resin with accelerator with microwave at 350 W for 3 min on/3 min off/3 min on for 1 day (2 exchanges) and placed in 60°C oven for polymerization for 2 days. Ultra-thin 50–100 nm sections were collected using a Leica UC6 Ultramicrotome. TEM specimens were stained with Sato's lead (3 min)/4% Uranyl Acetate in 70% methanol (4 min)/Sato's lead (6 min) prior to imaging on a FEI Technai BioTwin at 80 kV equipped with a Gatan UltraScan 1000 digital camera.

## RNAi via dsRNA feeding

RNAi feedings were performed as described previously (*Gurley et al., 2008*; *Rink et al., 2009*). Feeding and amputation schedules were tailored for each experiment and described in detail as following:

 *Figure 3C*: 5 dsRNA feedings (3 days in between).

 *Figure 4C,D*, *Figure 4—figure supplement 2A*: 8 dsRNA feedings (3 days in between).

 *Figure 4E,F*: 9 dsRNA feedings (3 days in between).

 *Figure 4—figure supplement 2B*, *Figure 4—figure supplement 3*: 6 dsRNA feedings (3 days in between) prior to amputation.

 *Figure 5A–C*: 3 dsRNA feedings (3 days in between).

*Figure 6*, *Figure 6—figure supplement 1C–E*, *Figure 6—figure supplement 2C*, *Figure 6—figure supplement 3*: 2 dsRNA feedings (3 days in between).

*Figure 7A–C*: 3 dsRNA feedings (2 days in between).

*Figure 7D–I*, *Figure 7—figure supplement 1A*: *IFT88-* and *LRRC50(RNAi)*: 3 dsRNA feedings (2 days in between); *DNAHβ-1(RNAi)*: 8 dsRNA feedings (2 days in between).

## Acknowledgements

We thank Jeffrey Lange and Aurimas Gumbrys for the planarian immobilization method using microfluidic devices, Jeffrey Lange and Zulin Yu for live imaging assistance, Eric Ross for bioinformatics assistance, and all member of the Sánchez lab, especially Li-chun Cheng and Sarah Elliott, for technical advice and critical suggestions. We also acknowledge all members of the Histology, Microscopy, Molecular Biology, and Reptile & Aquatics core facilities at the Stowers Institute for Medical Research for their technical support. This work was supported by NIH R37GM057260 to ASA. HTV is a Vietnam Education Foundation Fellow. ASA is an investigator of the Howard Hughes Medical Institute and an investigator of the Stowers Institute for Medical Research.

## Additional information

### Competing interests

ASA: Reviewing editor, *eLife.* The other authors declare that no competing interests exist.

### Funding

| Funder | Grant reference | Author |
| --- | --- | --- |
| National Institutes of Health (NIH) | R37GM057260 | Alejandro Sánchez Alvarado |
| Howard Hughes Medical Institute (HHMI) | Investigator | Alejandro Sánchez Alvarado |

The funders had no role in study design, data collection and interpretation, or the decision to submit the work for publication.

### Author contributions

HT-KV, Conception and design, Acquisition of data, Analysis and interpretation of data, Drafting or revising the article; JCR, ASA, Conception and design, Analysis and interpretation of data, Drafting or revising the article; SAMK, Analysis and interpretation of data; MMC, Acquisition of data, Analysis and interpretation of data; NL, Carried out phylogenetic analysis for the planarian solute carriers; RA, Performed STICS analysis

## Additional files

### Supplementary files

• Supplementary file 1. Summary information of the planarian homologs of *slc* genes.

• Supplementary file 2. Expression domains of *slc* genes in the planarian protonephridia.

• Supplementary file 3. Expression domains of *slc* genes in the rodent metanephros.

• Supplementary file 4. Summary information of the planarian homologs of human kidney disease genes.

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
