## [Decision Letter]

Thank you for sending your work entitled “Stem cells and fluid flow drive cyst formation in an invertebrate excretory organ” for consideration at *eLife*. Your article has been favorably evaluated by Janet Rossant (Senior editor) and three reviewers, one of whom, Yukiko Yamashita, is a member of our Board of Reviewing Editors. Another one of the reviewers, Friedhelm Hildebrandt, has also agreed to share his identity.

The Reviewing editor and the other reviewers discussed their comments before we reached this decision, and the Reviewing editor has assembled the following comments to help you prepare a revised submission.

In this manuscript, Alvarado and colleagues provide convincing evidence to show that the planarian excretory system, protonephridia, serves as a powerful model system to study mammalian kidney development and pathology such as cystic kidney disease. All reviewers agreed that this is an excellent paper, and raised no major concerns. They have several specific comments, which we suggest authors to address prior to publication. We decided to communicate all comments from reviewers to the authors, instead of summarizing points to be revised, as they contain encouraging comments and praises.

*Reviewer #1*:

This manuscript by Vu et al. provides a comprehensive characterization of the planarian excretory system. The authors conducted systematic phylogenetic and expression analyses of the ∼300 solute carrier (slc) transporters encoded by the planarian genome, identifying 49 that were expressed in the protonephridia. By mapping the *slc* expression patterns relative to known excretory system markers, the authors were able to define six different domains within this system. Strikingly, clustering *slc* genes based upon known substrate specificity and comparing their expression domains to those observed in vertebrate nephrons revealed a remarkable degree of similarity in the overall structural/functional organization of the planarian and rodent systems. These parallels extend to mutations associated with cystic kidney diseases in humans. RNAi knockdowns of planarian nephrins resulted in ultrastructural defects in the slit diaphragms in planarian flame cells and disruption of filtration. RNAi knockdowns of nephrocystin homologs generated cyst-like structures within the protonephridial tubules. Remarkably, the formation of these cysts and the resulting osmoregulatory defects are dependent upon proliferation. Finally, the authors show that knockdowns of genes required for proper ciliary beating (and thus, fluid flow within the ducts) also produced cysts within the excretory system.

*eLife* seeks to publish “authoritative” papers, and I can think of no better word to describe the truly impressive body of work that the authors present here. They convincingly show the many ways in which the planarian osmoregulatory system parallels such systems in vertebrates, providing an invertebrate model that can help us understand both the evolution of osmoregulatory systems and their pathologies. This beautiful paper sets the standard for how such work should be conducted in this organism and will help guide all future work on its excretory system.

To help strengthen the impact of this paper, I have only a few minor suggestions/comments that the authors may wish to consider addressing before publication:

1) The data presented in Figure 2 are so striking that they warrant two separate figures: one emphasizing the spatial domains defined by *slc* expression (A-H), and another figure emphasizing the structural/functional conservation with the rodent (J-L). In its current form, with the panels reduced to accommodate all of the data, these important conclusions are underemphasized.

2) Ultrastructural analysis of the *NPHS1* (RNAi) flame cell shows loss of the slit diaphragms, so the observed loss of ultrafiltration (500 kDa dextrans can now enter the system) seems counterintuitive. It may be helpful for the readers if the authors could comment on how loss of the slit diaphragms can lead to loss of size selectivity.

*Reviewer #2 (Friedhelm Hildebrandt)*:

This manuscript describes the protonephridial excretory system of *Schmidtea mediterranea* (planaria). In the first part of the manuscript, authors extensively characterized the planarian excretory system by identifying solute carrier (slc) genes and mapping their expression, and revealed that the planarian protonephridium is structurally and functionally similar to the vertebrate nephron. In the second part of the manuscript, authors knocked down planarian orthologues of human genes which, if mutated, cause kidney diseases including nephrotic syndrome and nephronophthisis. Authors showed by knocking down planarian *NPHS1* and *NEPH3* that planarian flame cells are functional homologues of vertebrate podocytes. Knockdown of planarian nephrocystin orthologues (*NPHP5*, *6* and *8*) led to development of cyst-like structures in proximal tubule segments. In addition, authors found that the cyst formation in planaria depends on overproliferation of protonephridial progenitor cells and loss of cilia function as a fluid flow generator/sensor. These have been also suggested as disease mechanisms of human cystic kidney diseases. Based on these findings, authors concluded that the planarian protonephridium not only has structural and functional homologies, but also shares pathologies with human nephron. In conclusion, authors suggest the planarian protonephridum as a model system to study human kidney diseases.

This work is interesting and the data are generally clear and of high quality. The amount of work to characterize the protonephridial excretory system of planarians is enormous, and extensive, and it is convincing and will have an impact in the renal community. The following points need to be addressed before publication.

1) The authors showed that cyst formation depends on increased proliferation of protonephridia progenitors upon knockdown of *NPHP8*. Have authors checked the increased proliferation in planaria with *NPHP5*, *NPHP6*, *IFT88*, *DNAHβ-1* or *LRRC6* knockdown?

2) The authors claim that fluid flow, which may be generated by cilia, is important for tubular function based on knockdown of motile cilia genes. However, whether the cilia-driven flow exists in planarian excretory system is not clear in the current data. Another possible scenario is that the motile cilia in the protonephridia may be important in clearing waste materials in a similar way to the ‘mucociliary clearance’ in the respiratory systems. Therefore, this review recommends that authors should tone the ‘cilia-driven fluid flow’ down in the Results and Discussion section.

3) It seems that knockdown of either NPNP genes or PCD genes in planaria resulted in protonephridial cysts, in other words, the same pathologies. However, in humans NPHP and PCD are totally different entities and almost mutually exclusive. In most cases, NPHP genes do not affect motile cilia and PCD genes do not affect immotile cilia. In this regard, it may be possible that the mechanism of cyst formation in planaria is different from that in human. In addition, in human NPHP, cysts occur in corticomedullary junctions and most develops from distal convoluted tubules. But protonephric cysts in planaria form in proximal tubules. Please discuss about these issues.

---

## [Author Response]

Reviewer #1:

*1) The data presented in*
Figure 2
*are so striking that they warrant two separate figures: one emphasizing the spatial domains defined by* slc *expression (A-H), and another figure emphasizing the structural/functional conservation with the rodent (J-L). In its current form, with the panels reduced to accommodate all of the data, these important conclusions are underemphasized*.

Thank you for this suggestion. We split the former Figure 2 into 2 independent figures and changed figure citations in the text accordingly.

*2) Ultrastructural analysis of the* NPHS1 *(RNAi) flame cell shows loss of the slit diaphragms, so the observed loss of ultrafiltration (500 kDa dextrans can now enter the system) seems counterintuitive. It may be helpful for the readers if the authors could comment on how loss of the slit diaphragms can lead to loss of size selectivity*.

The reviewer pointed out an important discrepancy between the EM data and the dextran assay, which has also been raised in studies of nephrotic syndrome in vertebrates. In fact, many nephrotic syndrome patients present severe proteinuria in conjunction with podocyte foot process effacement without observable podocyte detachment or loss (29; 54). We integrated a new paragraph in the paper explaining the current hypothesis in the field about how such big molecules could traverse the effaced podocyte foot processes that envelope the basement membrane:

“Why would the fusion of foot processes into a continuous sheet result in loss of filtration size selectively […], yet the exact mechanisms remain to be determined in both humans and planarians”.

Reviewer #2 (Friedhelm Hildebrandt):

*1) The authors showed that cyst formation depends on increased proliferation of protonephridia progenitors upon knockdown of* NPHP8*. Have authors checked the increased proliferation in planaria with* NPHP5*,* NPHP6*,* IFT88*,* DNAHβ-1 *or* LRRC6 *knockdown?*

We did check the increased proliferation in planarians after *NPHP5(RNAi)* (Figure 6), which, yielded a phenotype that was indistinguishable from *NPHP8(RNAi)* treated animals. We further added the analysis of *IFT88-* and *LRRC50(RNAi)* animals (Figure 7—figure supplement 2) and found again an increase in proliferation. Altogether, the data aptly support a general increase in proliferation upon both impairment of the NPHP complex and cilia function.

*2) The authors claim that fluid flow, which may be generated by cilia, is important for tubular function based on knockdown of motile cilia genes. However, whether the cilia-driven flow exists in planarian excretory system is not clear in the current data. Another possible scenario is that the motile cilia in the protonephridia may be important in clearing waste materials in a similar way to the ‘mucociliary clearance’ in the respiratory systems. Therefore, this review recommends that authors should tone the ‘cilia-driven fluid flow’ down in the Results and Discussion section*.

The reviewer raises an important point and we have therefore added more background information on how we reached this conclusion. Particularly, we included citations that have previously suggested fluid flow in protonephridia. For instance, Warner showed in his study of protonephridia in the rotifer *Asplanchna* that within 30 minutes after exposure to NiCl_2_, an inhibitor of cilia mobility, beating of cilia stopped and the bladder then ceased its normal fill-contraction cycle significantly, while other muscle and body movements appeared normal (101). Therefore, it is indeed likely that beating of the cilia is required to drive fluid flow into the protonephridial tubules. Additionally, cilia mobility has also been showed to be important for the generation of fluid flow into the pronephric tubules of zebrafish (52) and amphibians (107). The relevant section in the Discussion now reads:

“Planarians lack a circulatory system […] the direct observation of fluid flow cessation upon pharmacological inhibition of ciliary beating in the rotifer *Asplanchna* (73; 101).”

*3) It seems that knockdown of either NPNP genes or PCD genes in planaria resulted in protonephridial cysts, in other words, the same pathologies. However, in humans NPHP and PCD are totally different entities and almost mutually exclusive. In most cases, NPHP genes do not affect motile cilia and PCD genes do not affect immotile cilia. In this regard, it may be possible that the mechanism of cyst formation in planaria is different from that in human. In addition, in human NPHP, cysts occur in corticomedullary junctions and most develops from distal convoluted tubules. But protonephric cysts in planaria form in proximal tubules. Please discuss about these issues*.

We thank the reviewer for pointing out this aspect of our phenotypes. We have included a discussion paragraph comparing/contrasting the planarian phenotypes with human pathologies. This section now reads:

“Planarians have lost central components of centrosome duplication […] suggesting the likely existence of flow sensing role of motile cilia in planarian protonephridia (78; 92; 93; 94).”

“In NPHP knockdown planarians, cysts primarily arise in the PT at early stage […] why cysts predominantly affect the distal convoluted and collecting tubules only, but not other tubule parts in human NPHP is still unknown.”